# Large model structural uncertainty in global projections of urban heat waves

Zhonghua Zheng [1], Lei Zhao [1,2✉] & Keith W. Oleson [3]

Urban heat waves (UHWs) are strongly associated with socioeconomic impacts. Here, we use an urban climate emulator combined with large ensemble global climate simulations to show that, at the urban scale a large proportion of the variability results from the model structural uncertainty in projecting UHWs in the coming decades under climate change. Omission of this uncertainty would considerably underestimate the risk of UHW. Results show that, for cities in four high-stake regions – the Great Lakes of North America, Southern Europe, Central India, and North China – a virtually unlikely (0.01% probability) UHW projected by single-model ensembles is estimated by our model with probabilities of 23.73%, 4.24%, 1.56%, and 14.76% respectively in 2061–2070 under a high-emission scenario. Our findings suggest that for urban-scale extremes, policymakers and stakeholders will have to plan for larger uncertainties than what a single model predicts if decisions are informed based on urban climate simulations.

[1] Department of Civil and Environmental Engineering, University of Illinois at Urbana-Champaign, Urbana, IL, USA. [2] National Center for Supercomputing Applications, University of Illinois at Urbana-Champaign, Urbana, IL, USA. [3] Climate and Global Dynamics Laboratory, National Center for Atmospheric Research, Boulder, CO, USA. ✉email: leizhao@illinois.edu

Heat waves (HWs) – extremely high-temperature events – are among the most damaging climate extremes to human[1–5] and natural systems[6–9] globally. In the absence of effective adaptation or mitigation, extreme heat stress associated with climate change would cause a substantial increase in human mortality and morbidity[1,2,10], energy demand[11,12] and civil conflicts[13,14], and a large reduction in agricultural yield[15–18], livestock production[19], and workplace productivity[20]. In recent decades, HWs have been recognized as the deadliest environmental extreme in the United States (U.S.)[21,22]. These risks are further compounded in urban areas by the unique urban climates combined with concentrated population and assets[23]. Climate models agree on the projection of increasing severity, frequency, and duration of HWs at regional to global scales over this century under rising greenhouse gas emissions[24–27]. At a given grid cell or region, climate models do not necessarily agree with each other, but if all land grid cells are aggregated together, models agree remarkably[28]. These projections, however, are incapable of representing the HW signals for cities because: 1) the state-of-the-art Earth system models (ESMs) that participate in the Coupled Model Intercomparison Project (CMIP)[29,30] almost universally lack urban representation; and 2) the complex synergistic nature between urban heat islands and HWs[23,31] precludes the urban HW signal to be reduced to a simple anomaly on top of the traditional regional background climate projections by ESMs. Cities, as exposure hotspots for humans and infrastructure and fundamental foci of sustainable development and climate adaptation, need local-scale climate extreme projections that are specific to urban areas. Moreover, a multi-model urban framework is essential for managing the risks associated with the climate extremes, because urban planning and decision-making for enhanced resilience to extremes rely primarily on probabilistic estimates[32,33] which could only be obtained from the multi-model projections[34–36].

A robust modeling framework to address uncertainty in local- or regional-scale climate change should include the roles of internal variability (natural variability of the climate system resulting from nonlinear dynamical processes intrinsic to the atmosphere), structural uncertainty (uncertainty from choices in the climate model parameters, representation of unresolved physics and model design, and their effects on the climate sensitivity), and scenario uncertainty (uncertainty in prescribing future scenarios)[37]. Quantitative attribution of the uncertainty is particularly critical for assessing climate extremes, as the uncertainties in modeling the climate extremes are usually much larger than in modeling the mean climates. This uncertainty analysis has been done for the non-urban surfaces at regional scales using multi-modeled grid cell means[28,38]. For local-scale urban climate extremes, the role of internal variability can be addressed by rerunning the simulations a large number of times using a single urbanized climate model with small atmospheric initial condition perturbations[39]. However, quantification of the model structural uncertainty has never been achieved[39] because of the aforementioned near-universal lack of urban representation in ESMs. Nor could existing downscaling techniques (both dynamic and statistical) from a small number of global climate models or observational-based methods account for the full uncertainties associated with the extreme projections in cities on the global scale. This has been a critical research gap, as the structural uncertainty, in addition to scenario uncertainty, is expected to be the dominant source of uncertainty at the time horizons of multiple decades or longer[40].

Here in this work, we use a newly-developed urban climate emulator framework[41] to assess the inter-model variability in projections of local urban heat waves (UHWs) for the global urban areas in a future high-emission scenario, and to quantify the relative contributions of uncertainties from internal variability and model structural variability associated with the projections. The emulator framework combines process-based Earth system modeling using the U.S. National Center for Atmospheric Research's Community Earth System Model (CESM)[42] and the data-driven machine learning approach (see Methods). The internal variability is assessed based on the CESM Large Ensemble (CESM-LE)[43] simulations, whereas the model structural uncertainty is characterized based on the emulated multi-model projections. Note that the various CMIP5 ESMs that force the emulator likely have different choices for various climate model parameters, therefore the "structural uncertainty" evaluated in this study lumps together the uncertainties due to the model/parameterization design and due to the choices of parameters in the ESM. This study does not separate out the structural uncertainty and the climate model parametric uncertainty.

We find that the model structural uncertainty contributes substantially to the variability of multi-model projections of local-scale UHWs in the next several decades under climate change. Omission of the structural uncertainty would lead to a large underestimation of the risk of UHWs. We find that, for cities in four high-stake regions - the Great Lakes region of North America, Southern Europe, Central India, and North China - a gray swan UHW event with 0.01% likelihood projected by the CESM ensembles is estimated at the likelihood of 23.73%, 4.24%, 1.56%, and 14.76% respectively in 2061–2070 under a high-emission scenario by our model.

## Results and discussion
The urban climate emulator is built based on daily output from fully coupled simulations using the CESM. It incorporates all the atmospheric forcing fields that drive the urban land model in the coupled CESM as inputs and then outputs the urban daily temperatures. The urban model embedded in the CESM has been evaluated against both in situ and remote sensing observations over cities across the globe in previous studies[44–50]. It was further evaluated against PRISM (Parameter-elevation Relationships on Independent Slopes Model) observation-based climate data (http://www.prism.oregonstate.edu/) and the mesoscale dynamic downscaling results from ref. [51] over selected cities in the U.S.[41] Trained on fully coupled simulation outputs, the emulator is able to capture the dynamic land-atmosphere interactions in a CESM simulation statistically, including the feedback between urban ambient temperature and the anthropogenic energy use[52], because the impacts of these feedbacks have been preserved in the forcing and the urban response training sets (see Methods). We employ the tree-based XGBoost[53] to fit separate emulators for the present day (defined as 2006–2015) and future projected climate (2061–2070). The emulators are then applied to 17 ESMs that participated in the CMIP5[29] to generate global multi-model projections of local urban daily maximum ($T_{max}$) and minimum temperatures ($T_{min}$) under the Representative Concentration Pathway (RCP) 8.5 scenario. In this way, the emulator essentially functions by driving the urban model in CESM with atmospheric forcings from various ESMs in the CMIP5 in a statistical way instead of a numerical way (see Methods). Note that we use the same ten years of CESM simulations (2006–2015) to train the emulator for the present day, but thirty years (2051–2081) of data to train the emulator for the future period (2061–2070). This strategy aims to minimize the extrapolation errors associated with the machine learning when the emulator is applied to other CMIP5 ESMs (see Methods). We use a definition that has been shown to be related to the human mortality risk[1,5] to calculate UHWs for the present day (2006–2015) and future projected climate (2061–2070). Note that the "present-day" climate here is

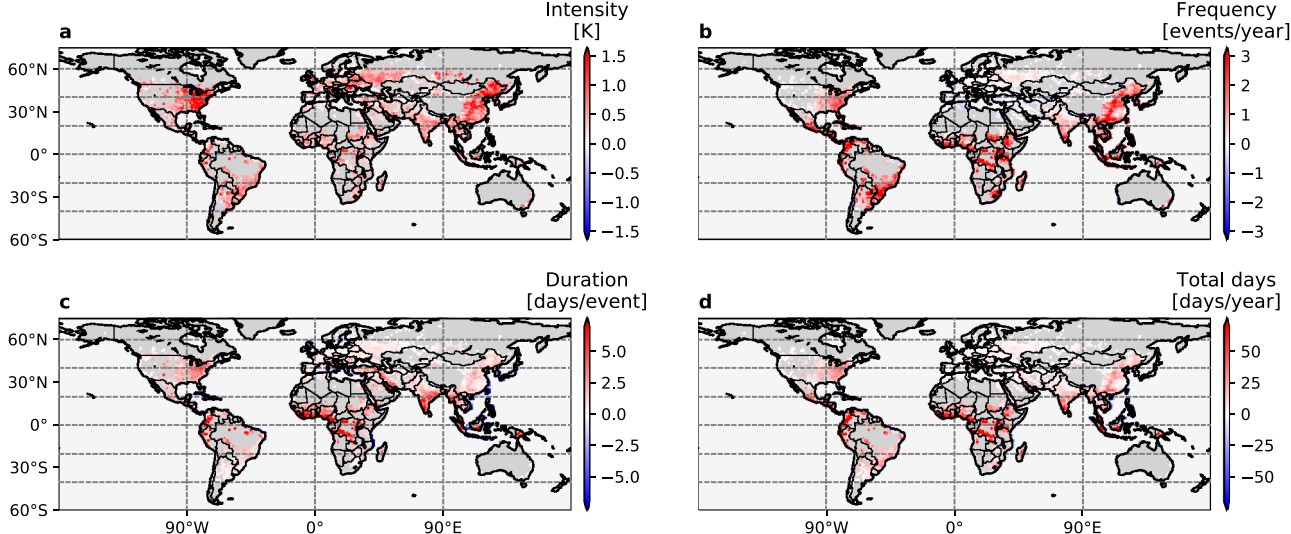

**Fig. 1 Difference of the multi-model ensemble mean change of the local urban heat waves and the background regional heat waves in 2061–2070 relative to 2006–2015 under RCP 8.5. a** intensity (K); **b** frequency (events per year); **c** duration (days per event); and **d** total days (days per year). Results are derived from the 17 selected Earth system models and the first member of the CESM-LE runs. Colors indicate grid cells that have urban land; and there are in total 4439 such grid cells. Dark gray and light gray indicate grid cells without urban land. Each colored point represents a decadal mean urban-minus-background difference in 2061–2070 relative to 2006–2015 within a 0.9° (lat.) × 1.25° (lon.) model grid cell.

technically a projection of "present-day" conditions because we are using the RCP 8.5 scenario only, rather than historical simulations. This is a reasonable assumption since the climate has largely followed the RCP 8.5 path in reality (instead of other paths such as RCP 4.5 and RCP 2.6). We document the results based on both $T_{max}$ (i.e., daytime) and $T_{min}$ (i.e., nighttime). Because their spatiotemporal patterns and thus conclusions are largely consistent with each other, we focus the discussion on the $T_{max}$ results in the main text. Results of $T_{min}$ are presented in the Supplementary Information.

**Future urban heat waves**. We show that the traditional projections from ESMs substantially underestimate the risks of UHWs in almost every aspect including intensity (average $T_{max}$ during the UHWs), frequency (average number of UHW events per year), duration (average number of days per UHW event) and total days (duration multiplied by frequency in days per year) (Fig. 1), due primarily to the neglect of urban physics. These positive anomalies relative to the background non-urban signals are not spatially uniform, confirming that the local-scale UHW projections cannot be simply reproduced by the traditional CMIP multi-model HW projections. Here, the "background non-urban" HW signals are based on the gridcell mean temperatures (i.e., the same as traditional climate projections), whereas the "urban" HW signals are based on the urban subgrid temperatures. According to the multi-model ensemble mean results, the increase in UHW intensity by 2061–2070 is underestimated by 1–2.6 K for 9.2% of the global urban areas compared to the traditional projections (Fig. 1a). These changes in UHW intensity are significantly larger than the average difference between the urban and background warming (−0.6 to 0.6 K)[41]. This indicates a more substantial underestimation of the projected risks in extreme conditions than in normal conditions by the traditional models for cities in a future warmer climate. The regions of large anomalies of change in UHW intensity generally colocate with those in frequency, duration and total days (Fig. 1), indicating further compounded underestimation of UHW-induced risk. In particular, the eastern U.S. and India are noteworthy as they are hotspots with large anomalies in all four aspects. Given massive urbanization expected to happen in India in the next few decades[54], the

underestimation of UHWs using traditional climate projections would put their city dwellers and infrastructure at a large risk.

Our multi-model results demonstrate the inter-model robustness of the increasing severity of UHWs over certain regions under climate change (Fig. 2 and Supplementary Fig. 6). CMIP models generally agree better in the projections of UHW intensity and frequency compared to duration and total days (Supplementary Fig. 1), as is also indicated by a greater spatial extent in the stippling shown in Fig. 2a, b than in Fig. 2c, d. This indicates that the structural uncertainty has a larger impact on the temporal distribution of daily temperature than on the magnitude. Models project similar magnitudes as well as the frequency of temperature extremes, but do not necessarily agree on which days of the year a heat wave event occurs. Note that the tropical (near-equator) region (15°S~15°N) is projected to have the most substantial increases in UHW frequency, duration, and total days by 2070. This is because the minimal seasonal variations in this region means that UHWs can occur at any time of the year (see Methods). Particularly susceptible to extreme warming (high-intensity increase) with high degree of inter-model agreement (high SNR) are four "hotspot" regions noteworthy: the Great Lakes region of North America, Southern Europe, Central India, and North China (Supplementary Fig. 2). Models agree, with high inter-model confidence, on the projection of substantial increases in the UHW intensity and frequency for cities in these regions in the next few decades under RCP 8.5 scenario (Fig. 2a, b), indicating large exposure of these cities to high extreme-heat risks. Specifically, the UHW intensity for cities in these regions is projected to increase by 2.2, 1.9, 1.4 and 2.0 K on average for the Great Lakes region of North America, Southern Europe, Central India, and North China respectively.

**Contribution of structural uncertainty**. Previous research has demonstrated a dominant role of the internal variability in the projections of hot extremes on the local and regional scales over the next few decades[28]. We show here that for urban areas, there is a large part of the uncertainties that result from the structural uncertainty. We define a structural uncertainty fraction (SUF) as the model structural uncertainty divided by the sum of internal variability plus model structural uncertainty to measure the

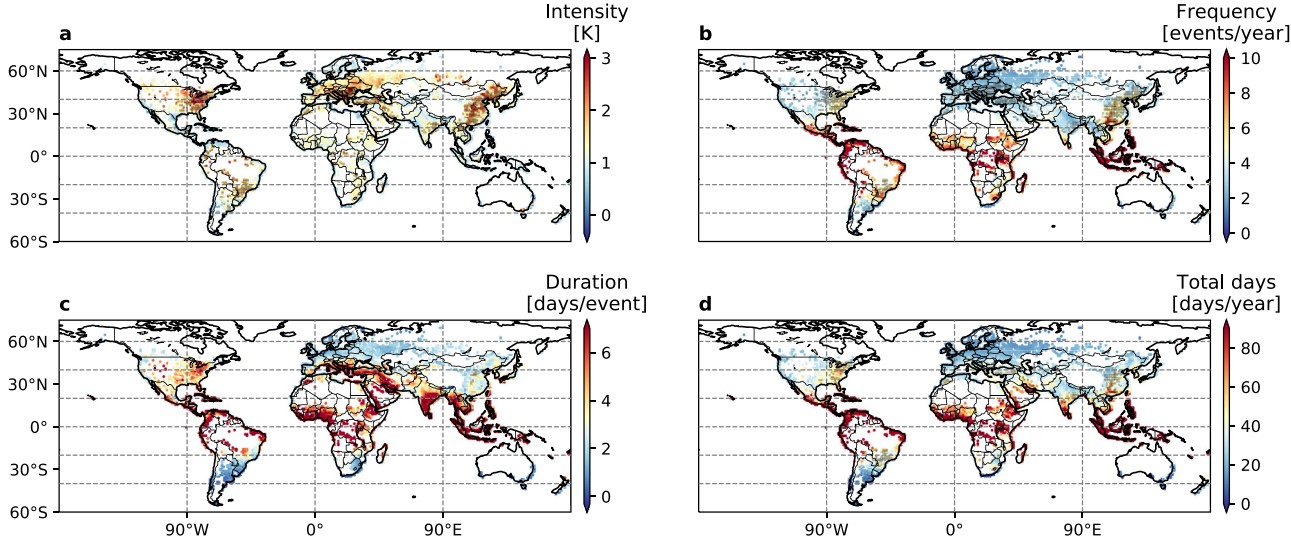

**Fig. 2 Global multi-model ensemble projections of urban heat wave changes in 2061–2070 relative to 2006–2015. a** intensity (K), **b** frequency (events per year), **c** duration (days per event), and **d** total days (days per year). Results are based on 17 selected Earth system models and the first member of the CESM-LE runs. Stippling indicates substantial change (intensity > 1.5 K) with high inter-model robustness (SNR > 2.0).

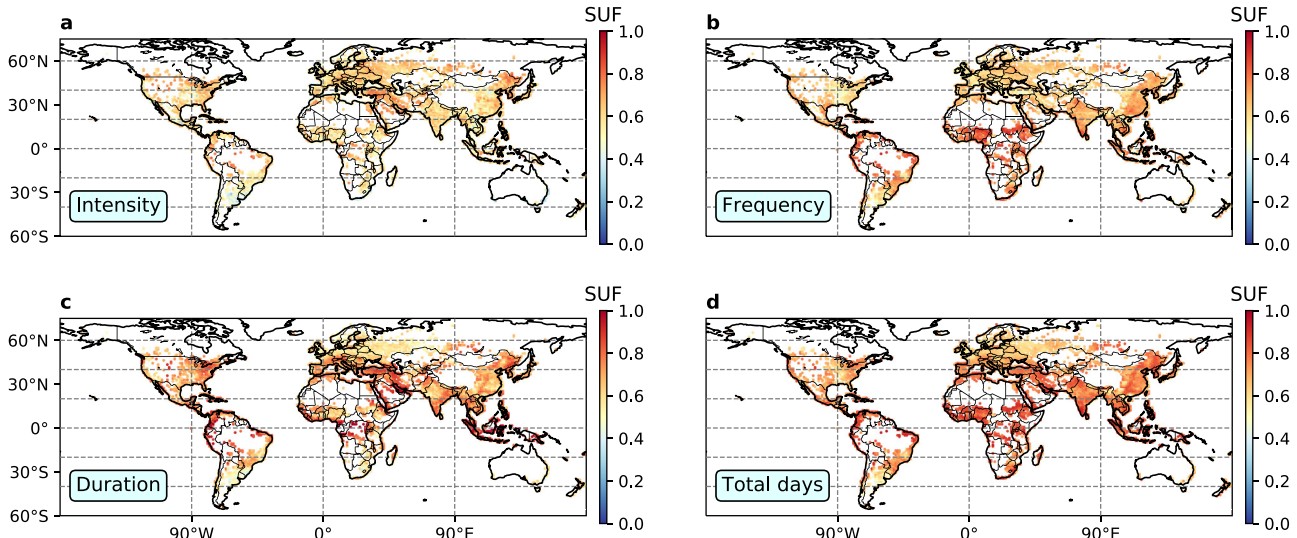

**Fig. 3 Relative contribution of the model structural variability in urban heat wave projections in 2061–2070 relative to 2006–2015 under RCP 8.5. a** intensity; **b** frequency; **c** duration; and **d** total days. Each colored point represents a decadal mean structural uncertainty fraction (SUF) defined as $\frac{\sigma_{CMIP}}{\sigma_{CMIP}+\sigma_{CESM}}$ within a 0.9° (lat.) × 1.25° (lon.) model grid cell. $\sigma_{CMIP}$ denotes the standard deviation across multi-model projections and $\sigma_{CESM}$ denotes the standard deviation across multi-member projections.

contribution of structural uncertainty. Based on the CESM-LE simulations and our emulated multi-model results, we find that the model structural uncertainty contributes more than 50% of the total variability by 2061–2070 for most of the global urban areas (Fig. 3 and Supplementary Fig. 7). The SUF with respect to projections in UHW frequency, duration and total days (Fig. 3b–d) are even larger than that in UHW intensity (Fig. 3a). These results indicate that for decision-making in the context of urban heat extremes based on climate modeled results, policy-makers and local practitioners might have to deal with the implications of large uncertainties in heat extremes on the local scale associated with the model structural spread. It is particularly important to account for this uncertainty where projections of future UHW frequency, duration and total days are concerned (Fig. 3b–d). Under the assumption that the structural uncertainty is primarily due to the existing climate model deficiencies, this part of the uncertainty driven by the multi-model spread of

atmospheric forcings could potentially be narrowed in the future development of ESMs with better-constrained climate model parameters and improved representations of physical and chemical processes. However, the progress of climate modeling convergence, despite the increased detail in representation of processes, may continue to remain slow[38].

The role of the model structural uncertainty in global UHW projections assessed in this study is essentially associated with larger-scale model structural design and parameter choices in various ESMs (such as radiative transfer, cloud microphysics, topography, dynamic land use land cover change, biogeochemical cycles, ocean model and atmospheric chemistry) rather than associated with the urban land scheme, because our emulator strategy uses various ESMs to drive a single urban model. In other words, the current emulator based on a wide number of ESMs characterizes the uncertainty in UHW projections due to larger-scale influences. Urban land (canopy) models generally represent

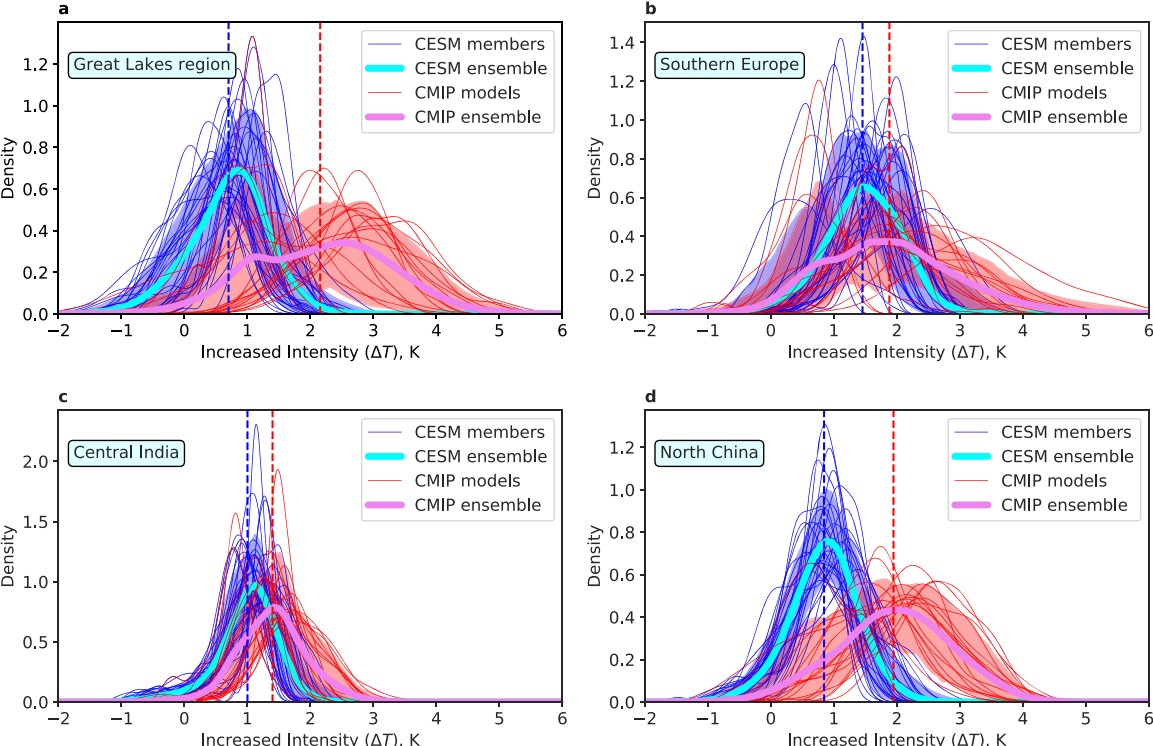

**Fig. 4 Probability distribution of changes in urban heat wave intensity by 2061–2070.** The thick lines mark the CMIP5 multi-model mean (thick violet lines) and CESM-LE multi-member mean (thick cyan lines). The thin red lines mark individual models of CMIP5; the thin blue lines mark individual CESM-LE members. The red and blue shading denote the 5th to 95th percentile across CMIP5 models and across CESM members respectively for each small bin. The changes are derived from land grid points in **a** Great Lakes region (36.15–49.5°N, 95–75°W), **b** Southern Europe (40–50°N, 15–30°E), **c** Central India (18–25°N, 75–87°E), and **d** North China (32–45°N, 110–123°E) between the 2006–2015 and 2061–2070.

less internal dynamics and feedbacks than the whole ESMs do, and thus would not drift far from one another if driven by an identical atmospheric forcing[55,56]. This is also evidenced by a comparison between two different urban land models over Contiguous U.S. urban areas in a recent study[41], which demonstrates markedly similar urban warming projections by forcing the two different urban land schemes (CLMU and WRF-Single Layer Urban Canopy Model) with the same atmospheric forcing. The magnitude of the uncertainty from large-scale climatology that the emulator addresses is much larger than the uncertainty introduced by different urban land parameterizations. The major variability in urban temperature projections is from the diverse larger-scale climate forcings projected by various ESMs (see Methods). We nevertheless acknowledge that it is advantageous to develop emulators from multiple urban land models that might be available in other ESMs in future to further assess the uncertainty associated with urban models.

**Gray swan urban heat waves.** An increasing number of record-breaking heat extremes has been observed recently over many cities globally[57–61], which raises the question of whether the probability of heat extremes on local scales is largely underestimated by the ESMs in a changing climate. We argue that this underestimation is likely for cities because of the lack of sampling a sufficiently wide uncertainty range. To illustrate the effect of accounting for the structural uncertainty on UHW projections, we calculate the probability density functions (PDFs) of the 10-year mean increase in UHW intensity (2061–2070 relative to 2006–2015) based on the emulated multi-model and CESM-LE urban results (see Methods). Results demonstrate a possible underestimation of heat extreme risks in cities if not accounting for the model structural spread (Fig. 4), as shown in illustrative

examples using the aforementioned four hotspot regions (Supplementary Fig. 2). The PDFs generated from multi-model urban projections cover much larger spectra than from the CESM multi-member projections which sample the internal variability only. This indicates that the probability of extreme increase in UHW intensity (higher tail in the PDFs) would potentially be misrepresented from single-model ensembles. As has been largely recognized, one of the major benefits of multi-model climate projections is the capability of capturing and quantifying all aspects of model uncertainties[62]. The above discussion demonstrates that the emulated multi-model results are unique in being able to account for the uncertainty associated with the model structural spread in local urban projections[41].

The underestimation of the urban extreme-heat risks shown above can be understood in the context of a "gray swan" event[63]. Unlike black swans, gray swans indicate the high-impact events that are not completely unanticipated. Here we define the "urban heat gray swans" as the heat extremes in cities that are virtually unpredictable based on historical observations or existing models but can be better foreseen and prepared for based on our model. We quantify the likelihood/probability associated with some urban heat gray swans to illustrate how plausible these events are in a changing climate. According to our results based on $T_{max}$, a seemingly unlikely urban heat gray swan event with 0.01% probability at any given year estimated by the CESM ensemble mean PDF is actually predicted, by the CMIP ensemble mean PDF, at the probabilities of 23.73%, 4.24%, 1.56% and 14.76% for cities in the four hotspot regions identified above (Great Lakes region of North America, Southern Europe, Central India, and North China), respectively. These results amount to that the once in 10,000 years urban heat gray swan event estimated previously using a single model are likely 4-, 24-, 64- and 7-year events. This

underestimation is even more concerning for cities in Central India (Fig. 4c) and North China (Fig. 4d), because the largest urban population growth is predominantly projected to concentrate in South and South East Asia and Africa by 2100[54].

Urban climate extremes often have very consequential socioeconomic implications, especially in future warmer climates. Plausible record-breaking heat extremes may not be predictable by extrapolating the sparse historical extremes in currently short urban historical records. Robust assessments of the risks associated with the UHWs under climate change have been limited by the near-universal absence of urban representation in the state-of-the-art ESMs. This limitation is overcome in our machine learning-enabled multi-model projections anchored in process-based simulations. We demonstrate a substantial contribution of the model structural uncertainty in projecting the local-scale urban climate extremes. Continuing model development including choosing better-constrained climate model parameters and representing more comprehensive dynamic processes in greater detail could potentially reduce this part of uncertainty. We note here that the emulator framework in this study is based on a single urban model – CLMU – simply because of the lack of other ESMs in CMIP5 that have an urban representation. It is important to include urban parameterizations in future development of various ESMs to more robustly assess the modeling uncertainty, especially those associated with the choices of urban land model parameters and design. We stress the critical need for both multi-model and urban-specific information (which traditional CMIP projections misrepresent) for projecting local urban climate extremes. Our findings indicate that the single-model (e.g., CESM) urban projections or dynamic downscaling from a small number of climate models substantially undersample the full uncertainty and thus underestimate the likelihood of urban heat extremes. The results suggest that policymakers and stakeholders will have to plan for larger uncertainties (risks) than what a single model usually predicts. It would be risky to just act on single-modeled signals; instead, local urban actions and adaptation decisions should plan for a wide range of possibilities which could be derived from the multi-model urban projections. For cities where major exposures (population and infrastructure) are expected to occur, our emulated urban results provide essential probabilistic information for estimating the likelihood of future extremes for decision-making in the context of risk management and preparedness and adaptation to climate-driven hazards.

## Methods

**CESM and large ensemble**. The Community Earth System Model (CESM)[42], hosted at the National Center for Atmospheric Research (NCAR), is a fully coupled Earth system model that provides state-of-the-art dynamic simulations of the Earth's climate states. It consists of seven components including Atmosphere, Sea-ice, Land, River, Ocean, Land-ice, and a Coupler that exchanges fluxes between the components. The land-atmosphere interactions are represented in its land component - the Community Land Model (CLM)[64]. As a sub-model embedded in CLM, the urban land scheme (CLMU) provides a physically-based urban representation and parameterization. More details on the biophysics and hydrology represented in CLMU can be found in ref. [65]. A global urban surface dataset prescribing the thermal (e.g., heat capacity and thermal conductivity), radiative (e.g., albedo and emissivity), and morphological (e.g., building height to street width ratio, roof areal fraction, average building height, and pervious ground fraction) characteristics of urban facets for each grid cell that has an urban land unit (i.e., urban subgrid) is provided by ref. [66]. The urban spatial extent is derived from a global population density dataset at 1 km spatial resolution. The urban property data is compiled by synthesizing a variety of datasets including satellite products, a global database of tall buildings, local building codes data and other municipal documentation, and validated against imagery from Google Earth[66]. With the urban surface dataset and the prognostic forcing data provided by the Atmosphere component of the CESM, the CLMU produces various energy, mass and momentum flux and state variables for each urban land unit in the grid cells globally.

The CLMU captures the dynamic feedback between the anthropogenic heat due to space heating and air conditioning (HAC) and the ambient environment using an embedded building energy scheme. The impact of this feedback can be important for local urban temperatures, especially under climate change[52]. The CLMU's building energy scheme dynamically models the building HAC energy use and associated wasteheat in urban areas[65]. Specifically, the internal boundary conditions for roofs and walls are determined by an approximation of internal building temperature held between prescribed maximum and minimum temperatures. Building interior maximum and minimum thermostat settings are also prescribed in the global urban surface dataset provided by ref. [66]. The amount of energy required to be added to increase the interior building temperature to the minimum temperature and the amount of energy required to be removed from the building interior to reduce the interior building temperature to the maximum temperature are respectively designated as the space heating and air conditioning fluxes. The heat removed by air conditioning is added as wasteheat (sensible heat) to the canyon floor. Wasteheat from inefficiencies in the heating and air conditioning equipment and from energy lost in the conversion of primary energy sources to end-use energy is also added as sensible heat to the canyon floor.

We utilized the simulations from the CESM Large Ensemble Project (CESM-LE)[43] (http://www.cesm.ucar.edu/projects/community-projects/LENS/) to constitute the training data and to quantify the role of internal variability in urban heat wave projections. The CESM-LE dataset includes 40 ensemble members of climate simulations conducted at the resolution of 0.9° latitude × 1.25° longitude. These ensemble members were run with slightly different atmospheric initial conditions (small random perturbations of the order of $10^{-13}$) but identical model configuration and climate scenarios. In this study, we utilized 32 members of the CESM-LE ensemble simulations under the Representative Concentration Pathways (RCP) 8.5 scenario.

### Urban daily temperature emulator

*Emulator.* We develop an urban daily temperature emulator, using the framework described in ref. [41], that maps the atmospheric forcing meteorology to the subgrid urban responses. The emulator framework provides a globally consistent and coherent method to characterize the uncertainty in urban warming due to large-scale climate variability. Detailed information on this urban climate emulator framework can be found in ref. [41]. The previous emulator function used in the framework was based on the multiple linear regression. Here in this study, we use a higher-capacity nonlinear model to fit the emulator function in order to better capture the daily-scale variability. The urban daily temperature emulator is a location-dependent (grid cell) non-parametric model based on the atmospheric forcings and time. It incorporates the daily atmospheric forcing fields from the atmospheric component of the ESM and the month-of-year indicator as the emulator inputs, and then outputs the daily maximum temperature ($T_{max}$) and daily minimum temperature ($T_{min}$). Specifically, the daily temperature $T_{max}$ or $T_{min}$ for a particular grid cell that has urban land unit, $T_{lat,lon}$, is emulated as

$$T_{lat,lon} = f_{lat,lon}(\mathbf{AF}, m) \qquad (1)$$

where $T_{lat,lon}$ denotes the urban daily maximum or minimum temperature; $\mathbf{AF}$ is the vector of atmospheric forcings from the atmospheric component of the ESM; $m$ is a 12-dimension vector of the month-of-year indicator; lat and lon are the latitude and longitude specifying the geospatial information of the grid cell; $f_{lat,lon}$ denotes the location-dependent emulator function. The $\mathbf{AF}$ matrix contains all the atmospheric forcing fields that drive the CLM in a coupled CESM simulation including net shortwave radiation, net longwave radiation and precipitation (liquid and solid) at the surface, and atmospheric temperature, pressure, specific humidity, and wind speed (zonal and meridional) at the forcing height. These forcing variables are consistently available in other CMIP5 ESMs. The emulator function ($f_{lat,lon}$) is a unique nonlinear model for each grid cell that has an urban land unit. Therefore, the impacts of urban surface form on its specific response to forcing meteorology are implicitly embedded in these location-specific functions ($f_{lat,lon}$). We employ the XGBoost[53] (a scalable tree boosting method) to fit the emulator function $f_{lat,lon}$ for 2006–2015 and 2061–2070 separately. Therefore, the fitted functions $f_{lat,lon}$ are a set of decision trees which map daily atmospheric forcing fields to urban temperatures at given grid cells. One of the salient features of the XGBoost is that the tree-based approaches are inherently able to capture the interactions among the predicting variables. Therefore, the nonlinear interactive effects between the forcing variables and their seasonality are represented in the XGBoost models.

We employed a Bayesian Optimization with 5-fold cross validation[67] to search for an optimal combination of three key hyperparameters of XGBoost including the number of gradient boosted trees (n_estimators), the maximum tree depth for base learners (max_depth), and the boosting learning rate (learning_rate). Because our entire global daily dataset is huge and thus expensive for multifold cross validation, we randomly sampled 0.1% of the entire training dataset of all the grid cells that have an urban land unit for both 2006–2015 and 2051–2080 which comprises more than 100,000 data samples for the search. The search space consists of n_estimators (ranging from 10 to 600), max_depth (from 3 to 7) and learning_rate (from 0.01 to 1). Based on the average out-of-sample performance of the 5-fold cross validation after 128 iterations, the best combination from our search space is 576 (for n_estimators), 6 (for max_depth), and 0.088 (for learning_rate).

Because the XGBoost-based emulator takes all the input variables that the CLMU needs in the coupled CESM to solve the equations, the emulator is essentially a statistical "solver" of the system of equations in the CLMU. The difference is that the emulator "solves" the equations for the output variables "statistically" instead of "numerically". In this sense, the emulator functions are not empirical relationships between the predicted variable (e.g., weather stations) and some large-scale atmospheric state variables (e.g., GCM or reanalysis data) as traditional statistical downscaling techniques seek. Instead, the emulator tries to reproduce the dynamically modeled urban variables by the CLMU. Traditional statistical downscaling has long been limited by the omission of climate feedbacks and the relatively arbitrary choice of incomplete predictors that remains the subject of debate[68,69]. In contrast, the emulator method in this study is trained on fully coupled climate simulation results and ingests a complete set of variables required for the CLMU dynamic simulations that have preserved the resulting impacts of the dynamic interactions and climate feedbacks. Therefore, the emulator captures the physical mechanisms and climate system feedbacks represented in the physical model (including the aforementioned feedback between urban temperature and HAC use[65]). See ref. [41] for more discussion on the implications and limitations of this urban climate emulator framework.

*Training data*. We utilize the fully coupled CESM-LE simulations as the training sets to build the urban daily temperature emulator. We randomly sample 10% of the 2006–2015 simulation outputs and 3.3% of the 2051–2080 from each member of the CESM-LE simulations within each grid cell that has an urban land unit to train the emulator for 2006–2015 and for 2061–2070 respectively. Each member then contributes to the training dataset with the same weight. With this strategy, we ensure a sufficiently large size of training data (>3 times of using only one CESM-LE member data). Note that we selected a much longer period (30 years) of data to train the emulator for the future period (2061–2070). This attempts to avoid extrapolation for the inference stage when the emulator is applied to other CMIP5 ESMs. With a longer period of training data, the emulator could "see" a much wider range of the features (forcing fields) and label (urban temperature). Supplementary Fig. 3 demonstrates that the training set well captures the ranges of the atmospheric forcing fields in CMIP5 models.

*Validation*. The whole framework including the urban modeling in the CESM and the emulator itself has been thoroughly evaluated in ref. [41]. Specifically, the urban simulation by the CLMU was evaluated against both the gridded ground-based observation dataset PRISM (http://www.prism.oregonstate.edu/) and the meso-scale dynamic modeling data over the contiguous U.S.[51] as well as over European[70] and Australian[71] cities[41]. Results demonstrated the credibility and robustness of our emulator method. In addition, the CLMU has also been widely evaluated against both in situ and remote sensing observations over various sites across the globe in previous studies[44–50].

Here, we further evaluated the statistical robustness of the urban daily temperature emulator by cross-member validation using the data of 32 CESM-LE members under RCP 8.5 that were excluded from the training of the emulator. Results show that our emulator accurately predicts the local urban $T_{max}$ across the globe. The out-of-sample global average root-mean-square-error (RMSE) of the urban $T_{max}$ across the tested ensemble members (Supplementary Fig. 4) are 0.73 K (for 2006–2015) and 0.74 K (for 2061–2070). The "error" here is defined as the difference between the daily values dynamically modeled by the CESM ensemble member and the ones by the emulator using atmospheric forcings from the same CESM ensemble member. The RMSE also indicates the additional component of the error (uncertainty) that the emulator introduces to the analysis, because the emulator "approximates" what the actual CLMU would produce in a coupled model system. These numbers are much smaller than the standard deviation (~3.1 K) of the urban $T_{max}$ across the CESM-LE ensemble members. The RMSE tends to be smaller at the equatorial and tropical regions, and relatively larger at the mid to high latitudes. In particular, the central and eastern U.S. appear to have the largest errors (Supplementary Fig. 4). We also compared the emulated CESM multi-member mean urban daily warming (2061–2070 minus 2006–2015) with the CESM directly modeled multi-member result (Supplementary Fig. 5); the difference between the two is markedly smaller than the emulated multi-model mean change in UHW intensity (Fig. 4a). These results above confirm the robustness of the emulator with unseen atmospheric forcing data.

**Multi-model projections**. The CESM-modeled atmospheric forcing fields that were used to train the emulator can be consistently extracted from other ESMs in CMIP5. These CMIP5 RCP simulations have been tuned and constrained to reproduce reasonably well the observed climate records[29]. We applied the emulator to all other available ESMs in CMIP5 to produce multi-model global urban $T_{max}$ and $T_{min}$ projections over 2006–2015 and 2061–2070 under RCP 8.5. We included all the ESMs that have their archive of outputs on a daily basis available, and excluded the results available from the same models yet just at a lower spatial resolution. In the end, a total of 17 ESMs were selected under RCP8.5 (Supplementary Table 1). The first ensemble member simulation of each selected ESM was used.

It is reasonable to argue that the atmospheric forcings from other CMIP5 ESMs might not be consistent with the CESM, because those models do not have urban

representation in their simulations. However, because ESMs are normally run at a relatively coarse spatial resolution (e.g., 0.9° latitude × 1.25° longitude) and the urban land units occupy very small fractions in the grid cells, the atmospheric forcings from the coupled simulations with or without urban representation are nearly identical. The CESM runs with the urban land unit replaced by the bare soil in the grid cell (a common configuration of other CMIP5 ESMs without an urban representation to fold the urban areas in as bare soil) but otherwise identical to the CESM-LE RCP 8.5 setup show indiscernible differences in their atmospheric forcings (<0.3% including all forcing fields used in the emulator) from the original CESM-LE simulations[41].

Direct application of the emulator requires the grids of all other ESMs to align exactly with the CESM grid, because the emulator function, as described above, is location-dependent. Therefore, we regridded the needed atmospheric forcing fields of all 17 ESMs from their native grids to the CESM grid based on the Earth System Modeling Framework (ESMF) regridding using a Python package "xESMF"[72] with the "patch" interpolation method[73,74]. These regridded atmospheric forcings accompanied by the encoded month-of-year indicator were fed as inputs to the emulator to generate 17 global projections of urban $T_{max}$ and $T_{min}$. The final multi-model analyses were then based on these 17 emulated projections plus the original CESM-LE dynamic simulations.

**Heat wave definition and calculation**. There are multiple ways to define a heat wave (HW) event[75–77], each of which has its own merits and disadvantages for different communities[78]. Here we use a "relative" definition of HWs that has been shown to be closely related to human mortality risk[1]. Within each grid cell, we define an HW event as three or more consecutive days with $T_{max}$ (or $T_{min}$ for the nighttime case) higher than the 98th percentile of $T_{max}$ ($T_{min}$) for the 2006–2015 period at the same grid cell. Using this definition, we calculated the urban HWs (urban subgrid) and the background HWs (whole grid cell) for each of the grid cells that has an urban land unit for the present day (2006–2015) and a future climate (2061–2070) under RCP 8.5. Note that we technically use a projection of "present-day" condition based on the RCP 8.5 scenario rather than historical simulations. We then assessed the changes in HW intensity (the average of daily maximum temperature during the HWs), frequency (number of HW events per year), duration (average total days per HW event), and total days (total number of HW days per year) between 2061–2070 and 2006–2015.

**Inter-model robustness**. We use the "signal-to-noise" ratio (SNR) as a measure of the inter-model robustness to illustrate how different ESMs agree on the projections of urban HWs. The SNR is calculated as the ratio of the multi-model ensemble mean to the inter-models variability as

$$SNR_{lat,lon} = \frac{\mu_{lat,lon}}{\sigma_{lat,lon}} \tag{2}$$

where $\mu$ denotes the multi-model ensemble means (i.e., means of HW intensity, frequency, duration, and total days); $\sigma$ denotes the inter-model variability calculated as the standard deviation of the multi-model values; and the "lat,lon" pair denotes each grid cell that has an urban land unit in the CLM. The reciprocal of SNR is the multi-model variability normalized by the mean, therefore SNR also indicates the multi-model spread. A smaller SNR implies higher model spread or smaller signal change such as multi-model mean change of HW intensity, frequency, duration and total days. We quantified the SNR for each grid cell that has an urban land unit to assess the inter-model agreement. We use SNR > 2.0 to indicate "high" inter-model agreement in the multi-model urban HW projections. Note that this threshold is somewhat subjective and for illustration purpose only. Different thresholds can be chosen for different applications of urban extreme events research, tightening or relaxing the tolerance in model disagreement.

**Internal variability and model structural uncertainty**. The discrepancy between multi-member simulations and between multi-model projections are associated with internal variability and model structural variability respectively. The internal variability refers to the natural variability of the climate system, which comes from the nonlinear dynamical processes intrinsic to the Earth-Atmosphere system. The role of this unpredictable internal variability is usually estimated by running the same ESM a number of times with slightly perturbed initial conditions but otherwise identical model configuration. Here we use the CESM-LE simulations to quantify the internal variability associated with the UHW projections.

On the other hand, the model structural uncertainty refers to the uncertainty introduced by the choices of model design (and climate model parameters in this study) and their effects on the climate sensitivity, which is unlikely to be represented by perturbing the initial conditions using a single model no matter how wide a range of the initial conditions is chosen. The underlying reason is that it is fundamentally impractical to describe all the climate processes accurately within a single ESM, no matter how complex the model itself. Different ESMs could be designed and tuned for different specific applications and therefore have their own choices of what processes to include and at what level of complexity. The multi-model ensembles try to tackle this issue by combining a set of model simulations from structurally independent models. It usually exhibits the better skill, higher reliability, and consistency of the model forecast[79,80]. Here we use the

emulated multi-model urban temperatures to compute the multi-model ensemble mean urban HW projection and to estimate the model structural uncertainty associated with the projection. We compare the estimated internal variability and the model structural uncertainty to characterize their relative contributions. The uncertainty due to the climate model parameters and due to the model structural design respectively are not separated out in this study.

One aspect of the structural uncertainty not addressed by this study is that associated with the choice of design and parameters of the urban land schemes. It is advantageous to develop emulators from multiple urban models to better assess the uncertainty associated with the urban land models. Nevertheless, the magnitude of the model structural uncertainty estimated in this study is reasonable, because the urban parameterizations, unlike ESMs, do not have many internal dynamics, and as such cannot drift far from each other if forced by identical atmospheric forcing[55,56]. Therefore, the major variability is from the diverse large-scale atmospheric forcings that are produced by different ESMs. The multi-model structural spread presented here is likely an upper bound encompassing the potential variability resulting from different urban parameterizations.

**PDFs of increase in UHW intensity**. We aggregated the changes of UHW intensity between 2061–2070 and 2006–2015 for the four hotspot regions (i.e., the Great Lakes region of North America, Southern Europe, Central India, and North China), and estimated their spatial probability density functions (PDFs) using a non-parametric method (Gaussian kernels) for both CESM-LE multi-member simulations and our emulated CMIP5 multi-model results. The Python package "SciPy"[81] was used to determine the kernel density for the PDF estimation.

## Data availability

All data used in this study is available at the UCAR/NCAR Climate Data Gateway via https://doi.org/10.5065/d6j101d1 and the CMIP5 Archive via https://esgf-node.llnl.gov/projects/cmip5/. The output data from the emulator are available in the public repository "Illinois Data Bank" via https://doi.org/10.13012/B2IDB-6081052_V1. Source data are provided with this paper.

## Code availability

Scripts and instruction[82] to develop and apply the urban climate emulator, and analyze the urban heat waves are available at https://doi.org/10.5281/zenodo.3872519 or https://github.com/zzheng93/code_uhws.

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

## Acknowledgements

We would like to acknowledge high-performance computing support from Cheyenne (https://doi.org/10.5065/D6RX99HX) provided by NCAR's Computational and Information Systems Laboratory, sponsored by the National Science Foundation. This work is based upon material supported by the NCAR, which is a major facility sponsored by the NSF under Cooperative Agreement No. 1852977. We thank AWS for providing AWS Cloud Credits for Research. L.Z. acknowledges the financial support from the Start-up Grant from University of Illinois, Urbana-Champaign.

## Author contributions

L.Z. designed the research; Z.Z. and L.Z. developed the emulator; Z.Z. performed data analysis with contributions from L.Z. and K.O.; Z.Z. and L.Z. drafted the manuscript, with discussions and contributions from K.O.; all authors reviewed the manuscript.

## Competing interests

The authors declare no competing interests.
