## [Peer Review File · Nature Communications]

Reviewer comments, first round –

Reviewer #1 (Remarks to the Author):

Review of NCOMMS-20-29834-T: "Large model parameter and structural uncertainties in global projections of urban heat waves" by Zheng et al.

Summary: The goal of the paper is to assess the effects of climate model uncertainty and internal variability on climate projections of urban heat waves. For their analysis, the authors train statistical emulators on responses from the Community Land Model with an urban scheme (CLMU) coupled to the Community Earth System Model (CESM) and then drive the emulator using fields generated from RCP 8.5 experiments in CMIP5 and the CESM Large Ensemble. The paper is well written, easy to follow, and the main results are impactful. Namely, the authors find that urban heat waves are more pronounced in climate models that represent urban effects, and that climate model uncertainty and internal variability are important to consider in future projections of urban heat waves. I am inclined to ultimately support the publication of this manuscript, however I also have a number of questions and suggestions that the authors should consider to improve its quality.

1: The emulation system applied in this manuscript appears to be described in more detail in another manuscript that is currently under review (ref 41, Zhao et al). While the authors provided a copy of this other paper to aid me in my review, I did not have time to scrutinize it carefully. The authors try to decouple the two papers by providing standalone information about the construction and validation of the emulators in the supplementary methods section. Taken at face value, the information in the supplementary section suggests that the emulators are reliable enough for their intended purpose. Nonetheless, if the reviewers of the other paper turn up an issue that jeopardizes the quality of this paper, the authors should disclose this information.

2: My main criticism of the approach is related to the lack of feedbacks between the emulator and the climate model forcing fields. The emulator takes in the forcing fields and computes a daily temperature maximum, but this temperature maximum does not feedback on the forcing fields. The land model embedded within CESM contains these feedbacks, while the emulator does not. One positive feedback that may be important for urban areas is between temperature and air conditioning systems (Takane et al., doi.org/10.1038/s41612-019-0096-2). As temperature increases, urban dwellers run their air conditioners more, which raises the temperature higher. Feedbacks between urban areas and the water vapor and energy budgets can also be important. Without these feedbacks, their estimates of model uncertainty on future urban heat waves may be overestimated or underestimated. I recommend that the authors attempt a rough assessment of the net magnitudes of the CLMU-atmosphere feedbacks relative to the uncertainty in their CMIP5 PDFs.

3: I agree that applying the emulator to CMIP5 captures a component of model structural uncertainty, but disagree that it also captures model parameter uncertainty. Model parameter uncertainty is usually assessed through perturbed parameter ensemble studies, which the authors are not including. While it is possible that embedding CLMU within a different model (e.g. MIROC5) may require small calibration adjustments of the CLMU parameters, which can be viewed as model parameter uncertainty, it is also likely that CLMU is not structurally compatible with other models. I suggest that the authors remove or reword the descriptions of model parameter uncertainty.

4: Using an emulator adds an additional component of error in the analysis, because the emulator is only an approximation of the actual CLMU system. The authors describe out-of-sample temperature RSMes of 0.6 K, which they compare to a temperature range of 2.0 K across the CESM LE members. This is a bit of an unfair comparison in my opinion. It would be better if they compared to the standard deviation across the realizations in the large ensemble, because to my eye the emulator error looks comparable to the widths of the shadings and sigmas of the CESM

PDFs in Figure 4.

5: I would like the authors to comment on the possibility to drive the CLMU directly at selected grid points using forcings from non-CESM models. If it is possible, why do they use emulators instead of CLMU? Can they use this technique to estimate emulator errors for non-CESM models? I am also curious about the computational savings in using emulators instead of the actual model. For other components of the CESM, emulators can provide enormous speed-ups, but perhaps the speed-ups are not as significant for CLMU.

6: To justify their uncertainty approach of using multi-model forcing fields to drive emulators based on a single model, the authors mention that differences across different urban parameterizations are likely to be small because urban models are linear. I do not fully agree with this claim. Even a set of different linear models can exhibit large differences, especially if the differences are amplified through climate feedbacks. I suggest the authors reword their justification on lines 134-138. The claim of linearity also contradicts the need for a non-linear XGBoost model to emulate the system, as described on lines 384-385.

7: I have a number of questions related to the emulators. The authors train their emulators on 10 years worth of present day and future conditions. Interannual trends and variability across the 10-year training periods can affect their predictions and may cause irreducible emulator errors. Are the authors accounting for interannual trends during training? If not, how large are the effects?

8: I am also wondering about the depth of the gradient boosted trees, learning rates, etc, which provide clues about the degree of non-linearity and interactions in the data. It would be useful if the authors included information in supplementary material about the values of the emulator hyperparameters and how they were estimated.

9: In terms of training and testing, the authors trained using 10% of data randomly selected from each CESM LE member, and tested on the held-out 90%. By training in this way, the emulators may have more error in generalizing to other models because they have seen data from each ensemble member. To improve generalizability, it may be better to use a different strategy. For instance, would the emulators perform better on other models if they were trained on 20% of the data from only 16 of the large ensemble members?

10: Other minor items are listed below.

* Line 47 suggests that model uncertainty is dominant at multi-decadal timescales, but scenario uncertainty related to future emissions and forcing is also critical.

* It would be useful to point out somewhere that there are other sources of internal variability in the climate system that are not considered in the CESM LE and your study (e.g. from the ocean state).

* The UHW durations in Figure 1 show large negative duration values in coastal western Pacific locations. Why does this occur?

* Can the authors confirm that none of the non-CESM models include any urban parameterizations so as to prevent double counting of urban effects?

* Compared to intensity and frequency, duration does not show many locations with significant changes in Figure 2. Perhaps the authors should move duration results to the supplementary section.

* Change 'mode' to 'model' at the beginning of line 125.

* On lines 497-502, the authors state, something to the effect, that it is impossible to produce larger responses by perturbing parameters in a single model than the spread across multiple models. This statement is not always true and should be altered. For certain quantities, I have seen larger responses from a perturbed parameter study of a single model than a multi-model ensemble. The reason is that the CMIP5 models have been tuned to produce reasonable climate

responses, but perturbed parameter ensembles set aside the tuning and can start to explore non-reasonable climates.

* In Figure 1, I recommend that the authors use a different color for the oceans because they have the same color as 0 on their colorbar. I also recommend that they cut out the high latitudes in their maps in all the figures to make it easier to see the differences and stippling.

Reviewer #2 (Remarks to the Author):

The manuscript entitled "Large model parameter and structural uncertainties in global projections of urban heat waves" by Zheng et al. submitted to Nature Communications quantifies the uncertainties in future heat waves in urban areas using CMIP5 simulations. The authors build an emulator to simulate urban temperature based on ensemble CESM runs. The same emulator is then applied to 17 ESMs, in which urban representation is absent. The intensity, frequency, duration, and total days of heat waves in urban areas using emulated urban temperatures are all greater than background regional values, consistent with existing findings. The authors also quantify the relative contribution of the internal variability and model parameter & structural variability. They found that CESM members slightly underestimate the increased intensity when compared to CMIP ensembles. Considering the increasing intensity and duration of heat waves in the near future, this study is meaningful, and the overall objective is quite attractive.

However, considering the potential flaws in its methodology, some conclusions of this study might not be valid. In particular, the urban daily temperature emulator is built upon the outputs of the ensemble CESM simulations, in which the dynamic (urban) land-atmosphere interactions are considered. But the 17 ESMs do not have urban land surface models. Such extrapolation of the trained emulator will inevitably involve uncertainties in the final "urban" projections, and more importantly, such uncertainties are almost unable to be quantified without the coupled "ESM-urban" simulations. This kind of uncertainties may even be greater for heat extremes. In particular, this may also be one of the reasons for the observed large structural uncertainty fraction (Fig. 3).

My second concern is that the model internal variability is quantified using ensemble CESM runs, while the model structural uncertainty is quantified based on 17 ESMs (each has only its "first" ensemble member), such that the two parts of the uncertainties are from two groups of models. The relative contribution based on such quantification may not be meaningful. A more ideal option would be quantifying the internal variability of all 17 ESMs (but the same urban emulator is still involved).

Here are some additional comments:

1. Abstract: The term "urban heat waves" should be either defined or replaced, as it is not a term commonly used in the field. The excessive heat in cities during heat waves is primarily caused by the mesoscale weather systems.
2. Abstract: "Our findings highlight that for urban-scale extremes, decision-makers and stakeholders have to accurately account for the multi-model uncertainties if decisions are informed based on climate simulations." However, the discussion of its implications remains rather superficial, and it is still not clear how "decision-makers and stakeholders" can use the results, especially considering the complexity of emulating urban signals using ensemble simulations.
3. Lines 122–125: "Under the assumption that the model parameter and structural uncertainty is primarily due to current model deficiencies, this part of the uncertainty could potentially be narrowed in future development of ESMs with better constrained mode parameters and improved representations of physical and chemical processes, ..." Again, the emulator is purely trained based on CESM simulations and its fidelity is not quantified when used in other ESMs. Under such assumption, whether the uncertainty can be narrowed or not may actually depend on the contribution of the dynamic urban surface-atmosphere interactions. (Same for lines 127–132).
4. Lines 147–149: "Results demonstrate considerable potential underestimation of heat extreme risks in cities if not accounting for model structural spread (Fig. 4)," – such statement is based on the fact that CESM-LE ensembles have relatively lower intensity estimates than the emulated CMIP5 ensembles. Picking a different ESM as the benchmark may yield a systematic overestimate

of intensity. (Same for lines 152–153 and the discussion on urban heat grey swan events.)
5. Figure 1: please point out whether this figure is the results of 17 emulated ESMs, 17 ESMs + ensemble mean of CESM runs, or 17 ESMs + 32 CESM runs.

Reviewer #3 (Remarks to the Author):

The authors aim to quantify uncertainty associated with modelled temperature extremes occurring in cities globally. If I understand correctly their main finding is that an ensemble consisting of members belonging to the same model (but each with slightly different initial conditions) undersamples temperature extremes compared to the situation where use is made of an ensemble composed of members belonging to different (CMIP5) models.

My main general comments are the following:

* I found the manuscript difficult to read, especially the 'Main' section. Often the language is inaccurate and sentences require multiple re-readings before they become clear, which adversely affects the fluency of the manuscript. Already the title is rather confusing in my opinion; it took me a while to grasp the meaning of "Large model parameter and structural uncertainties" (large model...? large ... parameter? large ... uncertainties?). Unfortunately, the cumbersome reading clouds the scientific content of the paper, making it difficult to judge its validity. It took a long time before I could even understand the main point made by the authors. Note, though, that the text of the 'Methods' section (at least, up to line 487) is much better than that of the 'Main' section; the paper would hugely benefit from bringing up the level of writing of the 'Main' section to that of the 'Methods' section.

* The authors' main point is that a single-model ensemble undersamples temperature extremes compared to multi-model ensembles. However, one could also argue the reverse, i.e. that large multi-model ensembles come with a higher probability of containing poorly performing models(outliers) with spurious outcomes hence potentially exaggerated extremes. I haven't found any discussion of this in the manuscript.

* I am wondering why the manuscript focuses on daily temperature maxima, while for cities it is well known that the daily temperature minima are affected more.

Specific comments:

ABSTRACT

"cities in the four high-stake regions" - as if these four regions are by their very nature 'high-stake': I would omit the word 'the' in 'the four high-stake regions'

"a virtually unlikely (0.1% probability) UHW event is estimated by our model with probabilities of...": confusing: the 0.1% probability pertains to the current situation I presume? If so, that should be made explicit. Or does it refer to the probability in the single-model ensemble while the multi-model ensemble exhibits higher probabilities? Difficult to understand...

"decision-makers and stakeholders have to accurately account for the multi-model uncertainties if decisions are informed based on climate simulations." => isn't this a bit too much 'stating the obvious'? The sentence also appears awkward to me.

MAIN

l.14: 'Absent effective adaptation...' - should this not rather be 'In the absence of ...'?

l.19: "Climate models agree on the projection of increasing severity, frequency, and duration of

HWs at regional to global scales or from an aggregated spatial probability perspective over this century under rising greenhouse gas emissions." => I don't understand the portion of this sentence starting at 'or from an aggregated...'. Is the 'aggregated spatial probability perspective' to be seen in opposition to 'regional to global scales'?

l.37: "Quantitative attribution of the structure of these uncertainties..." => I don't understand this

l.47: "is expected to be the dominant uncertainty sources..." => sources -> source

l.61: the emulator only outputs the daily maximum temperature? This appears odd, as the largest urban heat island intensity (urban-rural temperature increment) occurs at night.

l.64: 'urban daily temperature' is mentioned (I presume this refers to daily average temperature) while just prior to this sentence the authors state that daily maximum temperature is used.

l.64: the authors should provide a justification for their exclusive focus on RCP8.5, i.e., why ignore the other RCP scenarios?

l.69: "we are using RCP 8.5 scenario only" - article ('the RCP...') missing?

l.76: "We show that traditional projections from ESMs substantially underestimate the risks of UHWs" => is this because of the neglect of urban physics (i.e., the fact that the UHI phenomenon is not accounted for) or do ESMs underestimate the previously mentioned heatwave indicators also in 'background non-urban' areas?

l.85: sentence starting with "These numbers..." is not clear

l.86: I believe 'temperature difference' would in this context be more appropriate than 'warming difference', as warming implies a temperature tendency

l.87: what do the authors understand by 'non-extreme conditions'?

l.88: "to face more concerning change"... I am somewhat puzzled by the use of the word 'concerning'

l.88: "The regions of ... generally collocates..." => 'collocate'

l.91: "with large anomaly" => 'anomalies' (?)

l.97: "This indicates that the model parameter and structural uncertainty have larger impacts on the distribution of daily temperature than on the magnitude." not clear to me - I suppose that 'distribution' refers to 'spatial distribution'? Or is it rather 'distribution in time'?

l.103: Please explain how the four hotspot regions have been selected. From the manuscript I understand this has been done based on a visual inspection of the maps - is that correct?

l.115: "interval variability"? this should be 'internal variability' I suppose? Also, I guess the SUF would rather indicate the uncertainty related to the structural/parameter uncertainty, but that is more a semantic question perhaps.

l.120: "policymakers and local practitioners will need to consider the large model parameter and structural uncertainty on the local scale": I understand the intention of the authors, but find it odd to write that policymakers and practitioners, most of which are not scientists, would be assumed to consider any model technicalities.

l.125: "mode parameters" => ? (should this be 'model parameters'?)

l.127: "The role ... is essentially those" => 'that' (?)

l.127: "The role of model parameter and structural uncertainty in global UHW projections assessed in this study is essentially those associated with larger-scale model structural design and parameter choices in various ESMs..." => awkward sentence

l.127-140: The point made by the authors (that the observed variability arises mainly from the impact of large-scale atmospheric behaviour rather than from the urban land schemes) can be made in a much more compact way; also, since the heat extremes are caused by large-scale atmospheric behaviour, I find it odd to refer to the resulting variability/uncertainty as 'model parameter and structural uncertainty'

l.141: "Increasing..." => 'An increasing' (missing articles occur throughout the manuscript, the authors should check this throughout)

l.145 & many instances in the text elsewhere: very often the expression 'model parameter and structural uncertainty' is used, the authors may want to find a more compact expression for this (already in the title)

l.158: I am not sure whether I find the 'grey swan' analogy convincing, but that may simply be related to my lack of poetic imagination (you may ignore this comment...)

l.164: does the "0.1% probability" refer to events occurring with a probability of 0.1% within a given year?

l.168: in relation to previous point, should the 10,000 years not rather be 1,000 years?

METHODS

I.478: "frequency" => correct spelling

I.500: "Each ESM is suitable for specific applications due to its merit and shortcoming." => ? what is the meaning/use of this?

I.509 repeats almost literally earlier text (see I.127)

FIGURES

Fig.1: "Colored indicates grid cells that..." => rephrase ('Colours etc...?')

Fig.4: the authors may want to modify the layout as currently it does not always allow to distinguish the thick lines clearly from the ensemble-member lines (especially the blue)

Response to reviews on *Nature Communications* Manuscript NCOMMS-20-29834-T “Large model parameter and structural uncertainties in global projections of urban heat waves”

We particularly thank the reviewers for their constructive comments which helped us improve the manuscript for submission, and appreciate the opportunity to address their concerns and questions below. We address all the concerns raised by the reviewers on a point-by-point basis. The reviewers’ original comments are marked in blue, and our point-by-point responses are indicated in black with the tracked-change in red, below in this document. Please note that, for the reviewers’ convenience, all the line numbers below indicate the line numbers in the tracked-change version of the manuscript.

Response to Reviewers’ comments:

Table of Contents

REVIEWER #1.....	1
REVIEWER #2.....	14
REVIEWER #3.....	19

Reviewer #1

Review of NCOMMS-20-29834-T: “Large model parameter and structural uncertainties in global projections of urban heat waves” by Zheng et al.

“Summary: The goal of the paper is to assess the effects of climate model uncertainty and internal variability on climate projections of urban heat waves. For their analysis, the authors train statistical emulators on responses from the Community Land Model with an urban scheme (CLMU) coupled to the Community Earth System Model (CESM) and then drive the emulator using fields generated from RCP 8.5 experiments in CMIP5 and the CESM Large Ensemble. The paper is well written, easy to follow, and the main results are impactful. Namely, the authors find that urban heat waves are more pronounced in climate models that represent urban effects, and that climate model uncertainty and internal variability are important to consider in future projections of urban heat waves. I am inclined to ultimately support the publication of this manuscript, however I also have a number of questions and suggestions that the authors should consider to improve its quality.”

Thank you.

1: “The emulation system applied in this manuscript appears to be described in more detail in another manuscript that is currently under review (ref 41, Zhao et al). While the authors provided a copy of this other paper to aid me in my review, I did not have time to scrutinize it carefully. The authors try to decouple the two papers by providing standalone information about the construction and validation of the emulators in the supplementary methods section. Taken at face value, the information in the supplementary section suggests that the emulators are reliable enough for their intended purpose. Nonetheless, if the reviewers of the other paper turn up an issue that jeopardizes the quality of this paper, the authors should disclose this information.”

We thank the reviewers for raising this point. The other manuscript (ref 41, Zhao et al., 2020) that we submitted to accompany this current submission was the revised version after responding to its reviews. The revision addressed all the reviewers’ concerns and comments, some of which are actually also raised by the reviewers in this submission. The other manuscript (Zhao et al., 2020) has been accepted for publication and now in press by the journal. We intended to provide standalone information about the construction and validation of the emulator in this submission while trying to avoid repetition. However, the reviewer has raised a good point that readers may not be inclined to scrutinize both papers. Therefore, for the readers’ convenience, we have added some key methodological information that is described in Zhao et al. (2020) in this manuscript as well. This information will also help clarify some of the three reviewers’ other questions below. We have also modified the citation of Zhao et al. (2020) and provided its assigned DOI in this revision, however please note that this DOI might only be accessible after the publication date per *Nature Climate Change*’s policy.

2: “My main criticism of the approach is related to the lack of feedbacks between the emulator and the climate model forcing fields. The emulator takes in the forcing fields and computes a daily temperature maximum, but this temperature maximum does not feedback on the forcing fields. The land model embedded within CESM contains these feedbacks, while the emulator does not. One positive feedback that may be important for urban areas is between temperature and air conditioning systems (Takane et al., doi.org/10.1038/s41612-019-0096-2). As temperature increases, urban dwellers run their air conditioners more, which raises the temperature higher. Feedbacks between urban areas and the water vapor and energy budgets can also be important. Without these feedbacks, their estimates of model uncertainty on future urban heat waves may be overestimated or underestimated. I recommend that the authors attempt a rough assessment of the net magnitudes of the CLMU-atmosphere feedbacks relative to the uncertainty in their CMIP5 PDFs.”

The reviewer has raised a great point about the feedbacks embedded in CESM and the emulators. We appreciate the reviewer’s comment and the opportunity to clarify on this point. We have

discussed this in detail in Zhao et al. (2020). We agree with the reviewer that this is a critical aspect that we should also clarify in this manuscript.

The emulator method described in this study actually captures the dynamic feedbacks between the atmospheric forcing and land surface. The emulators are trained on the fully-coupled physical climate model simulations, and therefore preserve the physics represented in the physical downscaling model including the two-way land-atmospheric interactions and impacts of large-scale feedbacks and regional dynamics, as described in the original manuscript (Line 60 – 62). Please note that using atmospheric forcings from fully-coupled CESM run to drive its CLM is not equivalent to an “offline” simulation which lacks the land-atmosphere feedback. In the typical “offline” simulations such as CLM runs forced by site observations or reanalysis data, any changes of land surface state would not feedback to or change the state of the atmosphere. However, forcing CLM with the atmospheric forcing fields from the fully coupled CESM runs (with the same version of CLM), which is analogous to our emulator training, is actually a shortcut to retrieve the land surface state variables from the fully-coupled runs. It produces nearly identical land surface outputs to those from fully-coupled runs, because all the two-way land-atmosphere interactions represented in a fully-coupled configuration have been preserved in the forcing fields. Therefore, the emulators, trained on the atmospheric forcing and urban response outputs from fully-coupled simulations, have incorporated the land-atmosphere interactions such as the feedbacks between urban areas and the water vapor and energy budgets (as pointed by the reviewer).

In addition to these typical land-atmosphere interactions, the reviewer has also raised a good point on the positive feedback between urban temperature and air conditioning systems that we should clarify in the manuscript. We very much agree with the reviewer that this positive feedback could be important for urban areas. The CLMU (and hence the emulator) actually captures this feedback. The CLMU has an embedded building energy model to dynamically capture the anthropogenic heat release due to the space heating and air conditioning (HAC) in response to the urban ambient environment. This information and a brief description of the embedded building energy scheme have been noted in the original manuscript (Line 357 – 371). The modeled wasteheat fluxes of HAC are released into the urban canyon, interacting with canyon air temperature and thus with the atmospheric forcings through the surface fluxes. The “resulting” atmospheric forcing output and more importantly, the urban responses, from the coupled simulations has incorporated this feedback between urban air temperature and HAC. Therefore, trained on these atmospheric forcing fields as the predictors, our emulator also captures this two-way urban temperature – HAC feedback.

However, we do acknowledge that the atmospheric forcings from other CMIP5 ESMs without an urban parameterization do not capture the urban-atmosphere feedbacks in their atmospheric forcing fields. However, the effects of urban subgrid representation on grid cell-level atmospheric forcing variables are indiscernible as demonstrated by both our simulation (not shown) and previous studies (Hu et al., 2016; Zhang et al., 2016; Zhao et al., 2017), because the models are typically run at a relatively coarse spatial resolution (e.g. $0.9^\circ \text{ lat.} \times 1.25^\circ \text{ lon.}$) and the

urban areal fractions are very small in size compared to the grid cells (for reference, total urban area is about 2% of the Earth's land surface globally). These background atmospheric forcings that drive the CLMU in our emulator framework are analogous to those WRF-urban downscaling studies where the boundary conditions (usually provided by the “non-urbanized” GCMs or reanalysis data) that force the WRF-urban model are representative of a non-urban background atmosphere.

To demonstrate this point, we conducted two additional simulations identical to the CESM CMIP5 RCP 4.5 and RCP 8.5 runs from 2006 to 2100 but without urban land units represented (replaced with bare soil land unit in the grid cell). We compared the atmospheric forcings generated from these two simulations with those from the original CESM CMIP5 simulations (i.e. ones with urban land units), and found indiscernible differences (< 0.3% difference in the 50 atmospheric variables checked including all forcing fields used in the emulator). This comparison was reported in Zhao et al. (2020).

To address the reviewer's concern, we have now modified the following text in the Main section to improve the clarity:

“The urban climate emulator is built based on daily output from fully-coupled simulations using the CESM. It incorporates all the atmospheric forcing fields that drive the land model in the coupled CESM as inputs and then outputs the **urban daily temperatures**. **Trained on fully-coupled simulation outputs, the emulator is able to capture the dynamic land-atmosphere interactions in a CESM simulation statistically, including the feedback between urban ambient temperature and anthropogenic energy use (Takane et al., 2019), because the impacts of these feedbacks have been preserved in the forcing and the urban response training sets (see Methods)**. The emulator is then applied to 17 ESMs that participated in the CMIP5 (Taylor et al., 2012) to generate global multi-model projections of local **urban daily maximum temperature (T_{max}) and minimum temperatures (T_{min})** under the Representative Concentration Pathway (RCP) 8.5 scenarios. In this way, the emulator essentially functions by driving the urban model in CESM with atmospheric forcings from various ESMs in the CMIP5 in a statistical way instead of a **numerical way (see Methods)**.” (Line 60 – 70);

and in the Methods section:

“Because the XGBoost-based emulator takes all the input variables that the CLMU needs in the coupled CESM to solve the equations, the emulator is essentially a statistical “solver” of the system of equations in the CLMU. The difference is that the emulator **“solves” the equations for the output variables “statistically” instead of “numerically”**. In this sense, the emulator functions are not empirical relationships between **the predicted the predicted variable (e.g., weather stations) and some large-scale atmospheric state variables (e.g., GCM or reanalysis data) as traditional statistical downscaling techniques seek**. Instead, the emulator tries to reproduce the dynamically-modeled urban variables by the CLMU. Traditional statistical downscaling has long been limited by the

omission of climate feedbacks and the relatively arbitrary choice of incomplete predictors that remains the subject of debate (Fowler et al., 2007; Tang et al., 2016). In contrast, the emulator method in this study is trained on fully-coupled climate simulation results and ingests a complete set of variables required for the CLMU dynamic simulations that have preserved the resulting impacts of the dynamic interactions and climate feedbacks. Therefore, the emulator captures the physical mechanisms and climate system feedbacks represented in the physical model (including the aforementioned feedback between urban temperature and HAC use). See Zhao et al. (2020) for more discussion on the implications and limitations of this urban climate emulator framework.” (Line 444 – 458), and,

“It is reasonable to argue that the atmospheric forcings from other CMIP5 ESMs might not be consistent with the CESM, because those models do not have urban representation in their simulations. However, because ESMs are normally run at a relatively coarse spatial resolution (e.g. 0.9° latitude \times 1.25° longitude) and the urban land units occupy very small fractions in the grid cells, the atmospheric forcings from the coupled simulations with or without urban representation are nearly identical. The CESM runs with the urban land unit replaced by the bare soil in the grid cell (a common configuration of other CMIP5 ESMs without an urban representation to fold the urban areas in as bare soil) but otherwise identical to the CESM-LE RCP 8.5 setup show indiscernible differences in their atmospheric forcings ($< 0.3\%$ including all forcing fields used in the emulator) from the original CESM-LE simulations (Zhao et al., 2020).” (Line 499 – 507)

3: “I agree that applying the emulator to CMIP5 captures a component of model structural uncertainty, but disagree that it also captures model parameter uncertainty. Model parameter uncertainty is usually assessed through perturbed parameter ensemble studies, which the authors are not including. While it is possible that embedding CLMU within a different model (e.g. MIROC5) may require small calibration adjustments of the CLMU parameters, which can be viewed as model parameter uncertainty, it is also likely that CLMU is not structurally compatible with other models. I suggest that the authors remove or reword the descriptions of model parameter uncertainty.”

We appreciate the reviewer’s suggestion. The reviewer is correct that the model parameter uncertainty is usually assessed through perturbed parameter ensemble (PPE) approach which our study did not perform. In the original manuscript, we described the uncertainties as “model parameter and structural uncertainties” because the CMIP5 ESMs used to drive the emulator have different choices for various parameters (not the CLMU parameters). Therefore the variability derived from our multi-model emulation would lump both structural uncertainty and (climate) model parametric uncertainty together. To reflect this point, we have acknowledged in the original manuscript that this study does not separate the model parametric uncertainty and model structural uncertainty (Line 74 – 75). However, this (climate) model parameter uncertainty

could arguably be categorized into part of the structural uncertainty, and we agree with the reviewer that adding parameter uncertainty in the description might potentially lead to more confusion. Therefore, we have removed the text of “parameter uncertainty” throughout the manuscript, and noted a caveat to the readers that the “model structural uncertainty” in this study contains the uncertainty due to the climate model parameters. The text is reproduced below: “Note that the various CMIP5 ESMs that force the emulator likely have different choices for various climate model parameters, therefore the "structural uncertainty" evaluated in this study lumps together the uncertainties due to the model/parameterization design and due to the choices of parameters in the ESM. This study does not separate out the structural uncertainty and the climate model parametric uncertainty.” (Line 77 – 81)

We have also changed the title of the manuscript to “Large model structural uncertainty in global projections of urban heat waves.” This response would also help address Review 3’s Comment 1 below.

4: “Using an emulator adds an additional component of error in the analysis, because the emulator is only an approximation of the actual CLMU system. The authors describe out-of-sample temperature RSMEs of 0.6 K, which they compare to a temperature range of 2.0 K across the CESM LE members. This is a bit of an unfair comparison in my opinion. It would be better if they compared to the standard deviation across the realizations in the large ensemble, because to my eye the emulator error looks comparable to the widths of the shadings and sigmas of the CESM PDFs in Figure 4.”

This is a good point. We have calculated the standard deviation of urban daily maximum of average 2-m temperature across the realizations in the large ensemble. The averaged standard deviation is ~3.1 K for both 2006-2015 (3.1162 K) and 2061-2070 (3.1212 K). Please note that our previously reported urban daily temperature differences of ~2.0 K is not the maximum temperature range across the CESM LE members, but rather the global mean of the differences.

We also think that the reviewer has raised a good point about the “additional component of error” by the emulator that we should mention in the manuscript. The emulator tries to reproduce/approximate what CLMU simulates. The out-of-sample RMSE of the emulator indicates the additional component of error that the emulator introduces. This additional error is much smaller than the inter-member and inter-model variabilities. We have modified the text in the manuscript as below:

“The out-of-sample global average root-mean-square-error (RMSE) of the urban T_{\max} across the tested ensemble members (Supplementary Fig. 3) are 0.61 K (for 2006-2015) and 0.60 K (for 2061-2070). The “error” here is defined as the difference between the daily values dynamically modeled by the CESM ensemble member and the ones by the emulator using atmospheric forcings from the same CESM ensemble member. The RMSE also indicates the additional

component of the error (uncertainty) that the emulator introduces to the analysis, as the emulator “approximates” what the actual CLMU would produce in a coupled model system. These numbers are much smaller than the standard deviation (~ 3.1 K) of the T_{\max} across the CESM-LE ensemble members.” (Line 476 – 483)

5: “I would like the authors to comment on the possibility to drive the CLMU directly at selected grid points using forcings from non-CESM models. If it is possible, why do they use emulators instead of CLMU? Can they use this technique to estimate emulator errors for non-CESM models? I am also curious about the computational savings in using emulators instead of the actual model. For other components of the CESM, emulators can provide enormous speed-ups, but perhaps the speed-ups are not as significant for CLMU.”

This is a good question. Theoretically, it is possible to drive the CLMU dynamically at selected grid points using forcings from other non-CESM models. However, to dynamically run the CLMU would require hourly or at least 3-hourly atmospheric forcing fields rather than daily or monthly mean values, because the systems of equations (especially those partial derivative equations) in the model need to be solved at small temporal spacings by the numerical methods. The high temporal-frequency atmospheric outputs from the CMIP5 ESMs are extremely scarce in the CMIP archive. Therefore it is infeasible to directly force CLMU dynamically using the non-CESM models. However, the reviewer has pointed to an interesting future work if one wants to double check the errors for non-CESM models. One could repeat the CMIP RCP-scenario simulations using non-CESM models and archive the hourly output of atmospheric forcing variables (with great I/O and storage cost obviously though). With these forcings, dynamic CLMU simulations would then be possible.

In addition, it would be quite costly in terms of core-hours on the supercomputer to run CLMU forced by 17 hourly/3-hourly atmospheric datasets; whereas the emulator can be run on an analysis machine that is essentially free. To address the reviewer’s question regarding the computational savings, we provide some estimates as below:

- Emulator: One single core ~ 1 second to predict the 10 year (3650 days) daily urban temperatures for each urban grid cell;
- Actual CLMU: ~ 100 core-hours per simulated year for a 1-degree global offline simulation (would be even considerably higher for a fully-coupled simulation case).

6: “To justify their uncertainty approach of using multi-model forcing fields to drive emulators based on a single model, the authors mention that differences across different urban parameterizations are likely to be small because urban models are linear. I do not fully agree with this claim. Even a set of different linear models can exhibit large differences, especially if the differences are amplified through climate feedbacks. I suggest the authors reword their

justification on lines 134-138. The claim of linearity also contradicts the need for a non-linear XGBoost model to emulate the system, as described on lines 384-385.”

We appreciate the reviewer’s suggestion. The reviewer is correct that even a set of different linear models would exhibit large differences and hence nonlinearity, especially if the differences are amplified by climate system feedbacks. The point that we wanted to convey is that the magnitude of the variability (uncertainty) that the emulator addresses (from large-scale warming) is much larger than the variability (uncertainty) introduced by different urban canopy models which have weaker feedbacks than the climate systems (please see the comparison of CESM and WRF results for CONUS in Zhao et al. (2020) as one example). We did not mean to indicate urban models as “linear”. We have reworded the text as below:

“In other words, the current emulator based on a wide number of ESMs characterizes the uncertainty in UHW projections due to larger-scale influences. **Urban land (canopy) models generally represent less internal dynamics and feedbacks than the whole ESMs do, and thus would not drift far from one another if driven by an identical atmospheric forcing (Grimmond et al., 2010, 2011). This is also evidenced by a comparison between two different urban land models over the Contiguous U.S. urban areas in a recent study (Zhao et al., 2020), which demonstrates markedly similar urban warming projections by forcing the two different urban land schemes (CLMU and WRF-Single Layer Urban Canopy Model) with the same atmospheric forcing. The magnitude of the uncertainty from large-scale climatology that the emulator addresses is much larger than the uncertainty introduced by different urban land parameterizations. The major variability in urban temperature projections is from the diverse larger-scale climate forcings projected by various ESMs (see Methods).**” (Line 145 – 155)

7: “I have a number of questions related to the emulators. The authors train their emulators on 10 years worth of present day and future conditions. Interannual trends and variability across the 10-year training periods can affect their predictions and may cause irreducible emulator errors. Are the authors accounting for interannual trends during training? If not, how large are the effects?”

We thank the reviewer for the good question. One of the important practices in our emulator methodology is that when emulating on all other ESMs, the emulator should be applied to the same time range with the training period (Line 493 – 495) to avoid extrapolation. In two different time periods, the distribution of the data (magnitude, range and variance) could potentially be distinct due to climate change. Therefore, extrapolation (e.g. an emulator trained on 2006-2015 is used to predict 2061-2070) could introduce significant bias errors. In our other paper (Zhao et al., 2020), we trained the emulator over the entire period (2006-2100) of available monthly atmospheric forcings. In this study, because of the huge volume of daily atmospheric forcings, we trained the emulator over two separate periods 2006-2015 and 2061-2070 and then applied to the same time range of other ESMs. Within the 10-year periods, the inter-annual variability has been accounted for in the emulator because the raw data without any smoothing

are used in the training. We also had the month-of-year indicator as a feature in the emulator to capture the seasonal variability. The interannual variability in between the two periods (i.e. 2016 - 2060) was not captured in the emulator. However, this would not affect the emulation accuracy in the two training periods, as is demonstrated by the low root-mean-squared error (RMSE) and high coefficient of determination ($R^2 > 0.99$) of the two time periods.

The concern regarding the interannual trend usually lies in time-series prediction problems, where previous time steps of data are used to predict current and/or future steps. However, our emulator strategy is not a time-series prediction. At any given timestep (i.e. day), the atmospheric forcings of that step are used to predict urban temperatures; there is no information from previous time steps involved. The effect of interannual trend has already been represented in the atmospheric forcings. The emulator only resolves the dynamics from forcings to urban surface temperature.

8: “I am also wondering about the depth of the gradient boosted trees, learning rates, etc, which provide clues about the degree of non-linearity and interactions in the data. It would be useful if the authors included information in supplementary material about the values of the emulator hyperparameters and how they were estimated.”

This is a good suggestion and we appreciate the opportunity to provide the relevant information on hyperparameters. Although our location-specific emulator is trained separately at each grid cell, it is trained based on the same hyperparameters, because the emulators are essentially emulating the same CLMU over different grid cells. The impacts of different urban surface properties in different grid cells should be embedded in the location-specific XGBoost emulator function coefficients rather than the hyperparameters. We focused on the three key hyperparameters of XGBoost including the “number of gradient boosted trees (`n_estimators`)”, “maximum tree depth for base learners (`max_depth`)”, and “boosting learning rate (`learning_rate`)”. We used the Grid Search method with 10-fold cross validation to determine the best combination of hyperparameter values. Because our dataset is huge and thus very expensive for multifold cross validation, we randomly sampled 0.1% of the entire training data of all the grid cells for both 2006-2015 and 2061-2070 which ends up with more than 10^5 data samples to conduct the grid search. Each hyperparameter is tested with two values (low and high):

- `n_estimators`: 50, 500
- `max_depth`: 3, 6
- `learning_rate`: 0.05, 0.5

Based on the cross-validation, the algorithm concludes that the best combination is:

- `n_estimators`=500
- `max_depth`=6
- `learning_rate`=0.05

Following the reviewer’s suggestion, we have now added the information in the text (Line 435 – 443): “We employed a Grid Search with 10-fold cross validation to determine a best combination of three key hyperparameters of XGBoost including the number of gradient boosted trees (n_estimators), the maximum tree depth for base learners (max_depth), and the boosting learning rate (learning_rate). Because our entire global daily dataset is huge and thus expensive for multifold cross validation, we randomly sampled 0.1% of the entire training dataset of all the grid cells that have an urban land unit for both 2006-2015 and 2061-2070 which comprises more than 100 000 data samples for the grid search. The search space consists of n_estimators (50 and 500), max_depth (3 and 6) and learning_rate (0.05 and 0.5). Based on the average out-of-sample performance of the 10-fold cross validation, the best combination from our search space is 500 (for n_estimators), 6 (for max_depth), and 0.05 (for learning_rate).”

9: “In terms of training and testing, the authors trained using 10% of data randomly selected from each CESM LE member, and tested on the held-out 90%. By training in this way, the emulators may have more error in generalizing to other models because they have seen data from each ensemble member. To improve generalizability, it may be better to use a different strategy. For instance, would the emulators perform better on other models if they were trained on 20% of the data from only 16 of the large ensemble members?”

We thank the reviewer for the helpful suggestion. We have tested the strategy suggested by the reviewer (20% of the training data from 16 members) and compared with our original training. The results of emulator performance are shown in Table R1. In general, there is no significant improvement in the emulator out-of-sample performance found between the two training strategies.

Table R1. Emulator performance between two different training strategies.

Strategy	Time period	Averaged RMSE (K)	Averaged R ²
32 * 10% (strategy presented in paper)	2006-2015	0.6117	0.99485
16 * 20% (strategy presented by reviewer)	2006-2015	0.6137	0.99483
32 * 10% (strategy presented in paper)	2061-2070	0.5963	0.99481
16 * 20% (strategy presented by reviewer)	2061-2070	0.5977	0.99479

10: “Other minor items are listed below.

10.1 * Line 47 suggests that model uncertainty is dominant at multi-decadal timescales, but scenario uncertainty related to future emissions and forcing is also critical.”

Yes, thank you for pointing this out. We have updated the text (Line 47 – 49): “This has been a critical research gap, as the **structural uncertainty, in addition to scenario uncertainty**, is expected to be the **dominant source of uncertainty** at the time horizons of multiple decades or longer (Hawkins & Sutton, 2009).”

10.2: * It would be useful to point out somewhere that there are other sources of internal variability in the climate system that are not considered in the CESM LE and your study (e.g. from the ocean state).

We are not very clear on this comment. We think that some other sources of internal variability in the climate system (e.g. from the ocean state) are considered in the CESM-LE and hence our study. The CESM-LE ensemble members are fully-coupled simulations which means all the components of CESM are active and dynamically interact in the simulation including the Ocean model and Sea Ice model (not just land-atmosphere coupled). The perturbations to the initial conditions in each of the CESM-LE ensemble members include perturbations to the ocean state, therefore triggering the internal variability caused by ocean state. The forcing fields output from the CESM are the results after all the internal dynamics (variability) represented in the model have been simulated. Not only the two-way land-atmosphere interactions, but also the impacts of large-scale feedbacks (e.g. large-scale dynamics, ocean–air feedbacks, carbon climate feedbacks, and dynamic land use land cover change) and regional dynamics (e.g. the influence of topography on the regional atmospheric circulation and atmospheric rivers) represented in a fully-coupled configuration have been preserved in the atmospheric forcing variables.

10.3: * The UHW durations in Figure 1 show large negative duration values in coastal western Pacific locations. Why does this occur?

This is an interesting observation. Our results indeed show some interesting patterns of regional differences in UHW projections and their contrast to regional HWs. This regional variability is likely a combined effect of (i) regional background climatic effects, (ii) processes in urban surface energy balance, and (iii) different sensitivities of urban response to large-scale climate variability. The negative duration values in the coastal Western Pacific region in Figure 1 indicates less average duration days per one UHW event than per one regional HW event in these locations. Please note that except Duration (Fig. 1c), other metrics - Frequency (Fig. 1b) and Total days (Fig. 1d) - show positive (larger) anomalies compared to the regional HWs in this region. This indicates that the urban-regional difference is more relevant to the temporal distribution of high temperature

days between, i.e. more frequent shorter UHWs vs. less frequent longer regional HWs. We anticipate that the regional climate change and the biophysical processes in the urban surface energy balance in response to this change might play an important role here. However, further analyses and probably more control-experiment simulations in future work are needed to test this hypothesis.

10.4: * Can the authors confirm that none of the non-CESM models include any urban parameterizations so as to prevent double counting of urban effects?

Yes, the only other non-CESM model that has a “semi-urban” parameterization is the HadAM3 model developed by Met Office Hadley Centre has an urban representation (McCarthy et al., 2010, <https://doi.org/10.1029/2010GL042845>). The reason that it is a “semi-urban” land parameterization is that this model did not represent a physical urban surface or geometry, but rather implemented a typical vegetated/soil scheme with adjusted parameters better matching urban surfaces (Best, 2005; Best et al., 2006; McCarthy et al., 2010). This model was previously used to study the urban climate effects (McCarthy et al., 2010), but was not included in the CMIP5 simulations submitted by the Hadley Centre (HadGEM2-ES (Jones et al., 2011) and HadGEM2-CC (Martin et al., 2011)). Therefore, the urban effects were not double counted in our study.

10.5: * Compared to intensity and frequency, duration does not show many locations with significant changes in Figure 2. Perhaps the authors should move duration results to the supplementary section.

Thanks for the suggestion. We think that it is helpful to keep the duration results for completeness in the main text. This would also help the readers see how “total days” changes with “frequency” multiplied by “duration”. However, we would move the duration panels to the Supplementary Information, if the reviewer and the Editor concern the page limit issue for the Main text.

10.6: * Change ‘mode’ to ‘model’ at the beginning of line 125.

Done, thank you.

10.7: * On lines 497-502, the authors state, something to the effect, that it is impossible to produce larger responses by perturbing parameters in a single model than the spread across multiple models. This statement is not always true and should be altered. For certain quantities, I have seen larger responses from a perturbed parameter study of a single model than a multi-model ensemble. The reason is that the CMIP5 models have been tuned to produce reasonable climate responses, but perturbed parameter ensembles set aside the tuning and can start to explore non-reasonable climates.

This is an excellent point, and we highly appreciate the reviewer's suggestion. We actually did not intend to convey the inability to produce a larger spread by PPE using a single model than multi-models. The point we tried to make is that perturbing "initial conditions" (e.g. with very small random perturbations of the order of 10^{-13} in the CESM-LE; Line 400-401) in a single model (i.e. different ensemble members) is unlikely to represent the uncertainty caused by different choices of model design (i.e. multiple models). We have rephrased the text to reduce the misunderstanding as below (Line 551 – 559):

“On the other hand, the **model structural variability refers to the uncertainty introduced by the choices of model design (and climate model parameters in this study)** and their effects on the climate sensitivity, which is unlikely to be represented by perturbing the initial conditions using a single model no matter how **wide a range of the initial conditions is chosen**. The underlying reason is that it is fundamentally **impractical** to describe all the **climate processes accurately** within a single ESM, no matter how complex the model itself. **Different ESMs could be designed and tuned for different specific applications and therefore have their own choices of what processes to include and at what level of complexity**. The multi-model ensembles try to tackle this issue by combining a set of model simulations from structurally independent models.”

10.8: * In Figure 1, I recommend that the authors use a different color for the oceans because they have the same color as 0 on their colorbar. I also recommend that they cut out the high latitudes in their maps in all the figures to make it easier to see the differences and stippling.

Thanks for the good suggestion. Figure 1 and all other relevant figures have now been updated based on the reviewer's suggestion.

Reviewer #2

“The manuscript entitled “Large model parameter and structural uncertainties in global projections of urban heat waves” by Zheng et al. submitted to Nature Communications quantifies the uncertainties in future heat waves in urban areas using CMIP5 simulations. The authors build an emulator to simulate urban temperature based on ensemble CESM runs. The same emulator is then applied to 17 ESMs, in which urban representation is absent. The intensity, frequency, duration, and total days of heat waves in urban areas using emulated urban temperatures are all greater than background regional values, consistent with existing findings. The authors also quantify the relative contribution of the internal variability and model parameter & structural variability. They found that CESM members slightly underestimate the increased intensity when compared to CMIP ensembles. Considering the increasing intensity and duration of heat waves in the near future, this study is meaningful, and the overall objective is quite attractive.”

Thank you. We appreciate the positive comments on the contribution of our study.

1. “However, considering the potential flaws in its methodology, some conclusions of this study might not be valid. In particular, the urban daily temperature emulator is built upon the outputs of the ensemble CESM simulations, in which the dynamic (urban) land-atmosphere interactions are considered. But the 17 ESMs do not have urban land surface models. Such extrapolation of the trained emulator will inevitably involve uncertainties in the final “urban” projections, and more importantly, such uncertainties are almost unable to be quantified without the coupled “ESM-urban” simulations. This kind of uncertainties may even be greater for heat extremes. In particular, this may also be one of the reasons for the observed large structural uncertainty fraction (Fig. 3).”

This is a great question. Reviewer 1 has also raised the similar comment in Point 2. The reviewer is correct that the atmospheric forcings from other CMIP5 ESMs might not be consistent with CESM, because those models do not have urban representation in their simulations. We have tested the difference in atmospheric forcings between simulations with and without an urban land representation. The results were provided in Zhao et al. (2020). In summary, we compared the atmospheric forcings from the CESM CMIP5 simulations with those from our experiment CESM runs without urban land units but otherwise identical, and found indiscernible differences (< 0.3% difference in the 50 atmospheric variables checked including all forcing fields used in the emulator). The reason is that the urban areal fractions are very small in size compared to the whole grid cells (for reference, the total urban area is about 2% of the Earth’s land surface globally) and thus have minimum effects on the grid cell mean atmospheric forcings. To address

the reviewer's concerns, we have provided this information and added the clarification in the text (Line 499 – 507). Please see our response to Review 1's Point 2.

2. “My second concern is that the model internal variability is quantified using ensemble CESM runs, while the model structural uncertainty is quantified based on 17 ESMs (each has only its “first” ensemble member), such that the two parts of the uncertainties are from two groups of models. The relative contribution based on such quantification may not be meaningful. A more ideal option would be quantifying the internal variability of all 17 ESMs (but the same urban emulator is still involved).”

We thank the reviewer for the suggestion on “super-ensemble” in which all ensemble member simulations with different initial conditions for each climate model are incorporated in the multi-model ensemble computations (Tebaldi & Knutti, 2007). The other conventional way for constructing multi-model ensembles is to select one member from each model (Cattiaux et al., 2013; Tebaldi & Knutti, 2007), which is the approach used in this study. It actually remains the subject of debate which way is better. The number of ensemble member simulations for each ESM is considerably imbalanced in the CMIP. Therefore, taking all available members in the ensemble is essentially assigning higher weights to the models with more members because the differences between members of a single model are usually much smaller than differences between models. For example, averaging CESM of 40 ensembles with another model of 3 ensembles is similar to weighing CESM by 40 times while the other model by 3 times. Although this problem could be solved by weighting the individual members differently, this prior distribution is determined by human decisions and cannot be interpreted on a scientific basis (Tebaldi and Knutti, 2007). Imbalance in the number of ESM members induces extra complexity and uncertainty for the “weighting” for the evaluation.

We do agree with the reviewer that a more ideal option could be quantifying the internal variability of all the ensemble members (if available) of the 17 ESMs (emulator applied). However, there are not enough ensemble members for the other 17 ESMs in CMIP5. A number of CMIP5 ESMs only provide one member run, while other models have 3 to 5 member runs (<https://pcmdi.llnl.gov/mips/cmip5/>). Our study quantifies the internal variability and its relative contribution compared to structural variability based on the standard deviations. For each urban grid cell, calculating the standard deviations with only less than 5 samples is inadequate statistically. To the best of our knowledge, CESM is the only model in CMIP5 that provides a set of Medium Ensemble (15 members) and a set of Large Ensemble (40 members) simulations which can be used to compute statistically meaningful variance/standard deviation. Therefore, in this study we adopt the same strategy as in Fischer et al. (2013) where CESM is used to estimate internal variability and CMIP5 is used to estimate structural uncertainty.

In CMIP6, there will likely be more models that have larger ensembles. The reviewer has pointed to an interesting future work where the internal variability can be further assessed across a few models when their large-ensemble simulations become available.

“Here are some additional comments:

3. Abstract: The term “urban heat waves” should be either defined or replaced, as it is not a term commonly used in the field. The excessive heat in cities during heat waves is primarily caused by the mesoscale weather systems.”

Thanks. We have added a brief definition of urban heat waves in the abstract: “Urban heat waves (UHWs) — extreme heat events that occur in urban areas — are strongly associated with socioeconomic impacts.”

4. “Abstract: “Our findings highlight that for urban-scale extremes, decision-makers and stakeholders have to accurately account for the multi-model uncertainties if decisions are informed based on climate simulations.” However, the discussion of its implications remains rather superficial, and it is still not clear how “decision-makers and stakeholders” can use the results, especially considering the complexity of emulating urban signals using ensemble simulations.”

The point we wanted to convey here is that decision-makers and stakeholders should not make plans or decisions to adapt solely to one modeled signal (one model projection). Instead, adaptation and planning decisions should consider a wide range of possible outcomes. According to our results, one single model projection (even with a large number of ensemble numbers) would significantly underestimate the likelihood of an urban extreme heat event. However, using a single model (or a small number of regional downscaled models over limited regions) seems to be the only choice because multi-model urban projections were not available before. These findings have consequential policy implications.

In response to the reviewer’s concern, we have updated the text in the Abstract as:

“Our findings suggest that for urban-scale extremes, policymakers and stakeholders will have to plan for larger uncertainties (risks) than what a single model usually predicts if decisions are informed based on urban climate simulations,”

and elaborated in the discussion of Main text:

“We stress the critical need for both multi-model and urban-specific information (which traditional CMIP projections misrepresent) for projecting local urban climate extremes. Our findings indicate that the single-model (e.g., CESM) urban projections or dynamic downscaling from a small number of climate models substantially undersample the full uncertainty and thus underestimate the likelihood of urban heat extremes. The results suggest that policymakers and stakeholders will have to plan for larger uncertainties (risks) than what a single model usually

predicts. It would be risky to just act on single-modeled signals; instead, local urban actions and adaptation decisions should plan for a wide range of possibilities which could be derived from the multi-model urban projections. For cities where major exposures (population and infrastructure) are expected to occur, our emulated urban results provide essential probabilistic information for estimating the likelihood of future extremes for decision-making in the context of risk management and preparedness and adaptation to climate-driven hazards.” (Line 202 – 212)

5. “Lines 122–125: “Under the assumption that the model parameter and structural uncertainty is primarily due to current model deficiencies, this part of the uncertainty could potentially be narrowed in future development of ESMs with better constrained mode parameters and improved representations of physical and chemical processes, ...” Again, the emulator is purely trained based on CESM simulations and its fidelity is not quantified when used in other ESMs. Under such assumption, whether the uncertainty can be narrowed or not may actually depend on the contribution of the dynamic urban surface-atmosphere interactions. (Same for lines 127–132).”

The reviewer is correct that the emulator is based on one urban model and thus the results do not provide any information on whether the urban model-/emulator-induced uncertainty can be narrowed or not due to urban parameterization improvement. However, as we mentioned in the manuscript, the structural uncertainty in our study indicates the uncertainty from different designs of climate models rather than urban land models, because we use various ESMs to force the same CLMU. Therefore, “the uncertainty that could potentially be narrowed” here indicates the spread of atmospheric forcings produced by various ESMs that is potentially reducible with future better constrained and improved climate models. We agree with the reviewer that the current description might be ambiguous. To address the reviewer’s concern, we have now modified the text as below (Line 135 – 140):

“Under the assumption that the structural uncertainty is primarily due to the existing climate model deficiencies, this part of the uncertainty driven by the multi-model spread of atmospheric forcings could potentially be narrowed in future development of ESMs with better constrained climate model parameters and improved representations of physical and chemical processes. However, the progress of climate modeling convergence, despite the increased detail in representation of processes, may continue to remain slow (Knutti & Sedláček, 2013).”

As to the reviewer’s concern on urban surface-atmosphere interactions, please see our response to Point 1 and Reviewer 1’s Point 2.

6. “Lines 147–149: “Results demonstrate considerable potential underestimation of heat extreme risks in cities if not accounting for model structural spread (Fig. 4),” – such statement is based on the fact that CESM-LE ensembles have relatively lower intensity estimates than the emulated CMIP5 ensembles. Picking a different ESM as the benchmark may yield a systematic

overestimate of intensity. (Same for lines 152–153 and the discussion on urban heat grey swan events.)”

This is a good point. Our original argument is based on the fact that CMIP5 PDF curves (not just the ensemble mean one) cover a much larger range than CESM PDF curves (Line 167 – 168). We agree with the reviewer that picking a different ESM may yield an overestimated UHW intensity change if the ensemble average density curve is used. The reason lies in the “averaging” of multiple curves rather than the uncertainty/variability itself. Usually for extreme events analysis, all the curves/projections (e.g. max of the max) are used rather than the mean curve (e.g. max of the mean) to estimate the probability of extreme values, because “averaging” will mitigate the extremes.

In response to the reviewer’s concern, we have now noted the caveat in the text (Line 164 – 170):

“Results demonstrate a possible underestimation of heat extreme risks in cities if not accounting for the model structural spread (Fig. 4), as shown in illustrative examples using the aforementioned four hotspot regions (Supplementary Fig. 2). The PDFs generated from multi-model urban projections cover much larger spectra than from the CESM multi-member projections which sample the internal variability only. This indicates that the probability of extreme increase in UHW intensity (higher tail in the PDFs) would potentially be misrepresented from single-model ensembles.”

7. “Figure 1: please point out whether this figure is the results of 17 emulated ESMs, 17 ESMs + ensemble mean of CESM runs, or 17 ESMs + 32 CESM runs.”

Thank you for the good suggestion. In this figure, we treat CESM the same as other 17 ESMs, i.e. selecting the first ensemble member of CESM runs. We have now updated the captions of Figure 1: “Difference of the multi-model ensemble mean change of the local UHW and the background regional HW in 2061–2070 relative to 2006–2015 under RCP 8.5. a: HW intensity (K); b: frequency (events per year); c: duration (days per event); and d: total days (days per year). Results are derived from the 17 selected ESMs and the first member of the CESM-LE runs.”

Reviewer #3

“The authors aim to quantify uncertainty associated with modelled temperature extremes occurring in cities globally. If I understand correctly their main finding is that an ensemble consisting of members belonging to the same model (but each with slightly different initial conditions) undersamples temperature extremes compared to the situation where use is made of an ensemble composed of members belonging to different (CMIP5) models.

My main general comments are the following:

1. * I found the manuscript difficult to read, especially the 'Main' section. Often the language is inaccurate and sentences require multiple re-readings before they become clear, which adversely affects the fluency of the manuscript. Already the title is rather confusing in my opinion; it took me a while to grasp the meaning of "Large model parameter and structural uncertainties" (large model...? large ... parameter? large ... uncertainties?). Unfortunately, the cumbersome reading clouds the scientific content of the paper, making it difficult to judge its validity. It took a long time before I could even understand the main point made by the authors. Note, though, that the text of the 'Methods' section (at least, up to line 487) is much better than that of the 'Main' section; the paper would hugely benefit from bringing up the level of writing of the 'Main' section to that of the 'Methods' section.”

We appreciate the reviewer’s critique on the language of the manuscript. We are particularly grateful for the reviewer’s detailed editorial suggestions, which helps improve the presentation of our manuscript. As the reviewer has pointed out, the “model parameter and structural uncertainty” can be confusing. We have now changed the title and throughout the manuscript to keep it as “structural uncertainty”. We then define in the Main text that the structural uncertainty contains the parameter uncertainty associated with the ESMs. We have also modified the text based on the reviewer’s specific comments and incorporated the better language in the Methods section. We hope that the revised manuscript would address the reviewer’s language concerns.

2. * “The authors' main point is that a single-model ensemble undersamples temperature extremes compared to multi-model ensembles. However, one could also argue the reverse, i.e. that large multi-model ensembles come with a higher probability of containing poorly performing models(outliers) with spurious outcomes hence potentially exaggerated extremes. I haven't found any discussion of this in the manuscript.”

Thanks for raising this point. The reviewer is correct that large ensembles are possible to contain poorly performing models (outliers). However, such simulations are unlikely in CMIP5 RCP runs, because those RCP simulations in CMIP5 aim to provide reasonable future-scenario projections and therefore have been tuned/constrained to reproduce the observed historical climate reasonably well (Taylor et al., 2012). The outlier simulations or unreasonable climate responses are more likely to exist in other MIPs that are especially designed for testing the physics/chemistry in the models and the Perturbed Parameter Ensemble simulations as was also mentioned by Reviewer 1. Therefore, as demonstrated in other literature, multi-model ensemble method generally increases the skill, reliability, and consistency of model forecasts (Tebaldi & Knutti, 2007).

In response to the reviewer's concern, we have added the following text in the manuscript: "The CESM-modeled atmospheric forcing fields that were used to train the emulator can be consistently extracted from other ESMs in CMIP5. **These CMIP5 RCP simulations have been tuned and constrained to reproduce reasonably well the observed climate records (Taylor et al., 2012).**" (Line 491 - 493)

3. * "I am wondering why the manuscript focuses on daily temperature maxima, while for cities it is well known that the daily temperature minima are affected more."

We appreciate the reviewer's suggestion. The reasons that we focused on daily maximum temperature are: 1) most of the heat wave definitions, including the U.S. National Weather Service one, are based on the daily maximum temperature (daytime temperature); and 2) we have demonstrated in our other manuscript (Zhao et al. 2020) that, for the monthly means, the spatiotemporal pattern of diurnal minimum temperatures corresponds closely with diurnal maximums. Therefore we expect that in this study conclusions drawn from daily minimum temperatures would be similar to those drawn from daily maximums. However, the reviewer is correct that extreme high nighttime temperatures have been shown to be more closely associated with the mortality risk (Kusaka et al., 2012).

In response to the reviewer's question, we have built an emulator of urban daily minimum temperature using the same methodology and produced the emulated multi-model projections for urban daily minimum temperatures. We adopted the same definition of HW event using daily minimum temperature as using maximum temperature. Specifically, within each grid cell, an HW event is defined as three or more consecutive days with daily minimum temperature (T_{\min}) higher than 98th percentile of T_{\min} for the 2006-2015 period at the same grid cell.

As expected, the spatiotemporal patterns and conclusions drawn from T_{\min} largely follow the ones from T_{\max} . The graphs are reproduced below. These results have now been incorporated in the Supplementary Information and discussed in the Main text:

“This study does not separate out the structural uncertainty and the climate model parametric uncertainty. We document the results based on both T_{\max} (i.e. daytime) and T_{\min} (i.e. nighttime). Because their spatiotemporal patterns and thus conclusions are largely consistent with each other, we focus the discussion on the T_{\max} results in the main text. Results of T_{\min} are presented in the Supplementary Information.” (Line 80 – 84)

We have referred to these new figures based on T_{\min} in the appropriate locations in the text.

We also indicated in the Methods section that we built the emulators to project both urban T_{\max} and T_{\min} .

Figure R1. Multi-model ensemble mean change in average UHW (based on minimum temperature) intensity (K), frequency (events per year), duration (days per event), and total days (days per year) in 2061–2070 relative to 2006–2015 (17 ESMs and the first member of CESM runs). Stippling indicates substantial change (intensity > 1.5 K) with high inter-model robustness (SNR > 2.0).

Figure R2. Relative contribution of the model structural variability in UHW (based on minimum temperature) projections.

Figure R3. Spatial distribution of changes in urban heat waves (based on minimum temperature) intensity by 2061-2070.

Specific comments:

ABSTRACT

4. "cities in the four high-stake regions" - as if these four regions are by their very nature 'high-stake': I would omit the word 'the' in 'the four high-stake regions'"

Thank you. We have fixed it.

5. "'a virtually unlikely (0.1% probability) UHW event is estimated by our model with probabilities of...': confusing: the 0.1% probability pertains to the current situation I presume? If so, that should be made explicit. Or does it refer to the probability in the single-model ensemble while the multi-model ensemble exhibits higher probabilities? Difficult to understand..."

We are sorry that the "0.1%" is a typo, it should be "0.01%" (as documented in our code: https://github.com/zzheng93/code_uhws/blob/master/5_event_analysis/fig4_intensity.ipynb). This typo doesn't influence any other results. We have now corrected the typo in our manuscript. The "0.01% probability" refers to the probability estimated by the single-model ensemble. We have updated the text in the Abstract to clarify on this point:

"Results show that, for cities in **four high-stake regions** – the Great Lakes region of North America, Southern Europe, Central India, and North China – a virtually unlikely (**0.01%** probability) UHW event **projected by the single-model ensemble** is estimated by our model with probabilities of 13.91%, 5.49%, 2.78%, and 13.39% respectively in 2061–2070 under a high-emission scenario."

6. "decision-makers and stakeholders have to accurately account for the multi-model uncertainties if decisions are informed based on climate simulations." => isn't this a bit too much 'stating the obvious'? The sentence also appears awkward to me.

The text has now been modified as: "**Our findings suggest that for urban-scale extremes, policymakers and stakeholders will have to plan for larger uncertainties (risks) than what a single model usually predicts if decisions are informed based on urban climate simulations.**"

MAIN

7. (1.14): 'Absent effective adaptation...!' - should this not rather be 'In the absence of ...'?

Updated.

8. (1.19): "Climate models agree on the projection of increasing severity, frequency, and duration of HWs at regional to global scales or from an aggregated spatial probability perspective over this century under rising greenhouse gas emissions." => I don't understand the portion of this sentence starting at 'or from an aggregated...!'. Is the 'aggregated spatial probability perspective' to be seen in opposition to 'regional to global scales'?

The “aggregated spatial probability perspective” indicates the spatial PDF of the land fraction experiencing a certain change in extremes (or namely, spatial distribution of changes in extremes). It means that at a given grid cell or region, climate models do not necessarily agree with each other, but if all land grid cells aggregated together, models remarkably agree. This is the main point that the cited article (Fischer et al., 2013) right after the term “aggregated spatial probability perspective” discusses. In order to address the reviewer’s concern, we have now expanded the text as:

“Climate models agree on the projection of increasing severity, frequency, and duration of HWs at regional to global scales over this century under rising greenhouse gas emissions. At a given grid cell or region, climate models do not necessarily agree with each other, but if all land grid cells are aggregated together, models agree remarkably (Fischer et al., 2013).” (Line 19 – 22)

9. (1.37): "Quantitative attribution of the structure of these uncertainties..." => I don't understand this

The point we tried to convey here is that quantitative attribution of the uncertainties associated with extremes is critical, because the uncertainties of extremes are larger than the typical uncertainties of means. Understanding the uncertainties helps understand what portion of the uncertainty is reducible.

To address the reviewer’s concern, the sentence has been modified as: “Quantitative attribution of the uncertainty is particularly critical for assessing climate extremes, as the uncertainties in modeling the climate extremes are usually much larger than in modeling the mean climates.” (Line 37 – 39)

10. (1.47): "is expected to be the dominant uncertainty sources..." => sources -> source

We changed to “is expected to be the **dominant source of uncertainty**”.

11. (1.61): the emulator only outputs the daily maximum temperature? This appears odd, as the largest urban heat island intensity (urban-rural temperature increment) occurs at night.

We have produced the nighttime results in addition to the daytime ones. Please see our response to Point 3.

12. (1.64): 'urban daily temperature' is mentioned (I presume this refers to daily average temperature) while just prior to this sentence the authors state that daily maximum temperature is used.

Thank you for pointing it out. Because we have incorporated the daily minimum temperature results in response to the reviewer's previous comments, we have now specified the term as "urban daily maximum (T_{max})" and/or "daily minimum temperature (T_{min})" throughout the manuscript to clear the ambiguity.

13. (1.64): the authors should provide a justification for their exclusive focus on RCP8.5, i.e., why ignore the other RCP scenarios?

We provided the analysis of monthly results under both RCP 4.5 and RCP 8.5 in Zhao et al. (2020), and showed that results of RCP 4.5 largely follow the patterns of RCP 8.5 but just with a lower magnitude of urban warming (which is as expected). Therefore, in this study we focused on RCP 8.5 which our present climate trajectory has largely followed (instead of other RCPs) as we have pointed out in Line 74 – 75 in the original manuscript.

In addition, the CESM-LE project has only provided two scenarios: RCP 4.5 and RCP 8.5. The daily urban temperature data of the CESM-LE RCP 4.5 simulations are not available for training the emulator.

14. (1.69): "we are using RCP 8.5 scenario only" - article ('the RCP...') missing?

Added, thank you.

15. (1.76): "We show that traditional projections from ESMs substantially underestimate the risks of UHWs" => is this because of the neglect of urban physics (i.e., the fact that the UHI phenomenon is not accounted for) or do ESMs underestimate the previously mentioned heatwave indicators also in 'background non-urban' areas?

This is primarily because of the neglect of urban physics. The text has now been revised: "We show that **the** traditional projections from ESMs substantially underestimate the risks of UHWs in almost every aspect including intensity (average T_{max} during the UHWs), frequency (average number of UHW events per year), duration (average number of days per UHW event) and total days (duration multiplied by frequency in days per year) (Fig. 1), **due primarily to the neglect of urban physics.**" (Line 85 – 88)

16. (1.85): sentence starting with "These numbers..." is not clear

We have revised the sentence: "**These changes in UHW intensity** are significantly larger than the average **difference between the urban and background warming** (-0.6 to 0.6 K)." (Line 95 – 96)

17. (1.86): I believe 'temperature difference' would in this context be more appropriate than 'warming difference', as warming implies a temperature tendency

Yes. We indeed intend to indicate the temperature tendency (change) instead of absolute temperature here. Please see our revised sentence in response to Comment #16.

18. (1.87): what do the authors understand by 'non-extreme conditions'?

The term “non-extreme conditions” refers to the average. We have deleted “under the non-extreme conditions” in the manuscript. Please see our revised sentence in response to Comment #16.

19. (1.88): "to face more concerning change"... I am somewhat puzzled by the use of the word 'concerning'

The whole sentence has been changed to “**This indicates a more substantial underestimation of the projected risks in extreme conditions than in normal conditions by the traditional models for cities in a future warmer climate.**” (Line 96 – 98)

20. (1.88): "The regions of ... generally collocates..." => 'collocate'

Done.

21. (1.91): "with large anomaly" => 'anomalies' (?)

Done.

22. (1.97): "This indicates that the model parameter and structural uncertainty have larger impacts on the distribution of daily temperature than on the magnitude." not clear to me - I suppose that 'distribution' refers to 'spatial distribution'? Or is it rather 'distribution in time'?

Here it refers to temporal distribution (“distribution in time”). For a certain location, various climate models tend to agree on the intensity of the UHWs, but not on what days and how many days of the UHWs at a year. We have updated the manuscript accordingly: “This indicates that the **structural uncertainty** has a larger impact on the **temporal distribution** of daily temperature than on the magnitude.” (Line 108 – 109)

23. (1.103): Please explain how the four hotspot regions have been selected. From the manuscript I understand this has been done based on a visual inspection of the maps - is that correct?

Yes, this is correct. As we mentioned in the manuscript, the four hotspot regions are for illustrative purposes, and we do not imply these are the only hotspot regions. The four hotspot regions are selected based on their high values in intensity difference and high SNR values. We have revised the text: “**Particularly susceptible to extreme warming (high intensity increase) with high degree of inter-model agreement (high SNR) are four “hotspot” regions noteworthy:** the Great Lakes region of North America, Southern Europe, Central India, and North China (Supplementary Fig. 2).” (Line 113 – 116)

24. (l.115): "interval variability"? this should be 'internal variability' I suppose? Also, I guess the SUF would rather indicate the uncertainty related to the structural/parameter uncertainty, but that is more a semantic question perhaps.

Thank you, this has been corrected now. We have also removed the “parameter uncertainty” throughout the manuscript.

25. (l.120): "policymakers and local practitioners will need to consider the large model parameter and structural uncertainty on the local scale": I understand the intention of the authors, but find it odd to write that policymakers and practitioners, most of which are not scientists, would be assumed to consider any model technicalities.

This is a valid point. We have now updated the sentence: “These results indicate that for decision-making in the context of urban heat extremes based on climate modeled results, policymakers and local practitioners **might have to deal with the implications of large uncertainties in heat extremes on the local scale associated with the model structural spread.**” (Line 130 – 133)

26. (l.125): "mode parameters" => ? (should this be 'model parameters'?)

Thank you. Since we removed “model parameter uncertainties” from this manuscript, this typo has been fixed.

27. (l.127): "The role ... is essentially those" => 'that' (?)

We have removed the word “those” in (Line 141 – 145) to make the sentence more clear. Please see our revised sentence in response to Comment #28.

28. (l.127): "The role of model parameter and structural uncertainty in global UHW projections assessed in this study is essentially those associated with larger-scale model structural design and parameter choices in various ESMs..." => awkward sentence

We have revised the sentence (Line 141 – 145): “The role of **the model structural uncertainty** in global UHW projections assessed in this study is **essentially associated with** larger-scale model structural design in various ESMs (such as radiative transfer, cloud microphysics, topography, dynamic land use land cover change, biogeochemical cycles, ocean model and atmospheric chemistry) **rather than associated** with the urban land scheme, **because our emulator strategy uses various ESMs to drive a single urban model.**”

29. (l.127-140): The point made by the authors (that the observed variability arises mainly from the impact of large-scale atmospheric behaviour rather than from the urban land schemes) can be made in a much more compact way; also, since the heat extremes are caused by large-scale atmospheric behaviour, I find it odd to refer to the resulting variability/uncertainty as 'model parameter and structural uncertainty'

Please see our response above. The large-scale atmospheric behavior is caused by different choices of ESMs' model design which is now simply referred to as “structural uncertainty”.

30. (l.141): "Increasing..." => 'An increasing' (missing articles occur throughout the manuscript, the authors should check this throughout)

Done, thank you. We have also attempted to find and fix other instances of this in the manuscript.

31. (l.145) & many instances in the text elsewhere: very often the expression 'model parameter and structural uncertainty' is used, the authors may want to find a more compact expression for this (already in the title)

We have removed “model parameter uncertainties” throughout the manuscript. This is also suggested by Reviewer 1.

32. (l.158): I am not sure whether I find the 'grey swan' analogy convincing, but that may simply be related to my lack of poetic imagination (you may ignore this comment...)

Thanks. This concept is borrowed from Lin & Emanuel (2016): the original concept was used to describe the hurricane risks.

33. (l.164): does the "0.1% probability" refer to events occurring with a probability of 0.1% within a given year?

The value “0.1%” is a typo, it should be “0.01%”. Please see our response to Point 5. Yes, the “0.01% probability” refers to events occurring with a probability of 0.01% at a given year, and

thus indicating the event occurring once in 10 000 years (return period). This has been noted in the manuscript.

34. (1.168): in relation to previous point, should the 10,000 years not rather be 1,000 years?

Thank you. Please see our response above.

METHODS

35. (1.478): "frequency" => correct spelling

Corrected, thank you.

36. (1.500): "Each ESM is suitable for specific applications due to its merit and shortcoming."
=> ? what is the meaning/use of this?

The sentence has now been revised to improve the clarity as: "Different ESMs could be designed and tuned for different specific applications and therefore have their own choices of what processes to include and at what level of complexity." (Line 556 – 557)

37. (1.509) repeats almost literally earlier text (see 1.127)

The sentence has been modified to: "One aspect of the structural uncertainty not addressed by this study is that associated with the choice of design and parameters of the urban land schemes." (Line 565 – 566)

FIGURES

38. Fig.1: "Colored indicates grid cells that..." => rephrase ('Colours etc...?')

Thank you for the suggestion. We have updated the captions of Figure 1: "...Colours indicate grid cells that have urban land;..."

39. Fig.4: the authors may want to modify the layout as currently it does not always allow to distinguish the thick lines clearly from the ensemble-member lines (especially the blue)

The figure colour scheme has been updated per the reviewer's suggestion.

*****END*****

Reference:

Best, M. J. (2005). Representing urban areas within operational numerical weather prediction models. *Boundary-Layer Meteorology*, 114(1), 91–109.

- Best, M. J., Grimmond, C. S. B., & Villani, M. G. (2006). Evaluation of the Urban Tile in MOSES using Surface Energy Balance Observations. *Boundary-Layer Meteorology*, *118*(3), 503–525. <https://doi.org/10.1007/s10546-005-9025-5>
- Cattiaux, J., Douville, H., & Peings, Y. (2013). European temperatures in CMIP5: origins of present-day biases and future uncertainties. *Climate Dynamics*, *41*(11), 2889–2907. <https://doi.org/10.1007/s00382-013-1731-y>
- Fischer, E. M., Beyerle, U., & Knutti, R. (2013). Robust spatially aggregated projections of climate extremes. *Nature Climate Change*, *3*(12), 1033–1038. <https://doi.org/10.1038/nclimate2051>
- Fowler, H. J., Blenkinsop, S., & Tebaldi, C. (2007). Linking climate change modelling to impacts studies: recent advances in downscaling techniques for hydrological modelling. *International Journal of Climatology*, *27*(12), 1547–1578. <https://doi.org/10.1002/joc.1556>
- Grimmond, C. S. B., Blackett, M., Best, M. J., Barlow, J., Baik, J.-J., Belcher, S. E., et al. (2010). The International Urban Energy Balance Models Comparison Project: First Results from Phase 1. *Journal of Applied Meteorology and Climatology*, *49*(6), 1268–1292. <https://doi.org/10.1175/2010JAMC2354.1>
- Grimmond, C. S. B., Blackett, M., Best, M. J., Baik, J. J., Belcher, S. E., Beringer, J., et al. (2011). Initial results from Phase 2 of the international urban energy balance model comparison. *International Journal of Climatology*, *31*(2), 244–272. <https://doi.org/10.1002/joc.2227>

- Hawkins, E., & Sutton, R. (2009). The Potential to Narrow Uncertainty in Regional Climate Predictions. *Bulletin of the American Meteorological Society*, 90(8), 1095–1108.
<https://doi.org/10.1175/2009bams2607.1>
- Hu, A., Levis, S., Meehl, G. A., Han, W., Washington, W. M., Oleson, K. W., et al. (2016). Impact of solar panels on global climate. *Nature Climate Change*, 6(3), 290–294.
<https://doi.org/10.1038/nclimate2843>
- Jones, C. D., Hughes, J. K., Bellouin, N., Hardiman, S. C., Jones, G. S., Knight, J., et al. (2011). The HadGEM2-ES implementation of CMIP5 centennial simulations. *Geoscientific Model Development*, 4(3), 543–570. <https://doi.org/10.5194/gmd-4-543-2011>
- Knutti, R., & Sedláček, J. (2013). Robustness and uncertainties in the new CMIP5 climate model projections. *Nature Climate Change*, 3(4), 369–373.
<https://doi.org/10.1038/nclimate1716>
- Kusaka, H., Hara, M., & Takane, Y. (2012). Urban Climate Projection by the WRF Model at 3-km Horizontal Grid Increment: Dynamical Downscaling and Predicting Heat Stress in the 2070's August for Tokyo, Osaka, and Nagoya Metropolises. *Journal of the Meteorological Society of Japan*, 90b, 47–63. <https://doi.org/10.2151/jmsj.2012-B04>
- Lin, N., & Emanuel, K. (2016). Grey swan tropical cyclones. *Nature Climate Change*, 6(1), 106–111. <https://doi.org/10.1038/nclimate2777>
- Martin, G. M., Bellouin, N., Collins, W. J., Culverwell, I. D., Halloran, P. R., Hardiman, S. C., et al. (2011). The HadGEM2 family of Met Office Unified Model climate configurations. *Geoscientific Model Development*, 4(3), 723–757. <https://doi.org/10.5194/gmd-4-723-2011>

- McCarthy, M. P., Best, M. J., & Betts, R. A. (2010). Climate change in cities due to global warming and urban effects. *Geophysical Research Letters*, *37*(9).
<https://doi.org/10.1029/2010GL042845>
- Takane, Y., Kikegawa, Y., Hara, M., & Grimmond, C. S. B. (2019). Urban warming and future air-conditioning use in an Asian megacity: importance of positive feedback. *Npj Climate and Atmospheric Science*, *2*(1), 1–11. <https://doi.org/10.1038/s41612-019-0096-2>
- Tang, J., Niu, X., Wang, S., Gao, H., Wang, X., & Wu, J. (2016). Statistical downscaling and dynamical downscaling of regional climate in China: Present climate evaluations and future climate projections. *Journal of Geophysical Research: Atmospheres*, *121*(5), 2110–2129. <https://doi.org/10.1002/2015JD023977>
- Taylor, K. E., Stouffer, R. J., & Meehl, G. A. (2012). An Overview of CMIP5 and the Experiment Design. *Bulletin of the American Meteorological Society*, *93*(4), 485–498.
<https://doi.org/10.1175/Bams-D-11-00094.1>
- Tebaldi, C., & Knutti, R. (2007). The use of the multi-model ensemble in probabilistic climate projections. *Philosophical Transactions of the Royal Society A: Mathematical, Physical and Engineering Sciences*, *365*(1857), 2053–2075. <https://doi.org/10.1098/rsta.2007.2076>
- Zhang, J. C., Zhang, K., Liu, J. F., & Ban-Weiss, G. (2016). Revisiting the climate impacts of cool roofs around the globe using an Earth system model. *Environmental Research Letters*, *11*(8), 084014. <https://doi.org/10.1088/1748-9326/11/8/084014>
- Zhao, L., Lee, X., & Schultz, N. M. (2017). A wedge strategy for mitigation of urban warming in future climate scenarios. *Atmospheric Chemistry and Physics*, *17*(14), 9067–9080.
<https://doi.org/10.5194/acp-17-9067-2017>

Zhao, L., Oleson, K. W., Bou-Zeid, E., Krayenhoff, E. S., Bray, A., Zhu, Q., et al. (2020). Global multi-model projections of local urban climates. *Nature Climate Change (in Press)*.

<https://doi.org/10.1038/s41558-020-00958-8>

Reviewer comments, second round

Reviewer #2 (Remarks to the Author):

The authors have made a good effort in revising the manuscript, and they adequately addressed all my previous review. I have no further comments.

Reviewer #3 (Remarks to the Author):

I am happy with the way the authors have addressed my comments from the initial review.

Reviewer #4 (Remarks to the Author):

I was asked to fill in for Reviewer #1 of the first round of reviews. I feel that his/her points were overall addressed sufficiently and will not comment further on the replies to the other two reviewers. Instead, I would like to ask to the authors about a major additional concern I have concerning their emulation strategy:

The issue of extrapolation using the boosted tree emulator already came up several times during the first round of reviews, but never in a context that I would have expected to be most obvious: the application of an emulator trained on one model system (CESM) to another, with potentially completely different values ranges for the atmospheric forcings. This could/should be a major problem I am afraid, because it is well-known that tree models are not able to extrapolate. Even basic blog posts have discussed such issues with tree models to great length, e.g.

<http://freerangestats.info/blog/2016/12/10/extrapolation>

So, my question is how the emulator should at all be able to reproduce realistic temperature extremes (or their uncertainties) provided input of atmospheric forcings that could lie well outside its training range. For example, if the surface warming attained by 2070 is between 2.5 and 3.0 degrees across the CESM ensemble members, how would the emulator be able to predict a realistic heat extreme for another model which has warmed by only 2 degrees or even 4 degrees? Tree models should be particularly bad in these situations (even among machine learning models). The authors say in reply to Reviewer #1 on p.8 of the response to the reviewers that they have mitigated such effects by using data from the same decade for each CMIP model (i.e. the same decade as used to train CESM). However, CESM and the CMIP models will almost certainly have warmed to substantially different degrees (and of course this will likely also apply to other forcing variables). I expect that the CMIP model inputs ranges to the emulator will be substantially wider than the training range encountered in CESM. I recommend the authors demonstrate that this is not the case and otherwise check if their study is still valid/what the implications are.

Related to this point: I also find the range of hyperparameters optimized over not particularly comprehensive. I am further wondering why one would train on 50 estimators as well if you can afford 500, this computational expense would have been better spent on more detailed grid searches for other hyperparameters.

Minor comment:

- p.5, l.110: you mean 'on which day of year', or 'on which day in a series of days'? Otherwise, this really doesn't make sense to me

**Response to reviews on *Nature Communications* Manuscript NCOMMS-20-29834A
“Large model structural uncertainty in global projections of urban heat waves”**

We particularly thank Reviewer #4 for filling in for Reviewer #1’s role to review our first round of response to reviews. We appreciate Reviewer #4’s further comments and questions, and the opportunity to address the Reviewer’s concerns below. We provide our response to all the questions raised by the reviewer on a point-by-point basis. The reviewers’ original comments are marked in blue, and our point-by-point responses are indicated in black with the tracked-change in red, below in this document. Please note that, for the reviewer’s convenience, all the line numbers below indicate the line numbers in the tracked-change version of the manuscript.

REVIEWER COMMENTS

Reviewer #2 (Remarks to the Author):

1: “The authors have made a good effort in revising the manuscript, and they adequately addressed all my previous review. I have no further comments.”

Thank you.

Reviewer #3 (Remarks to the Author):

2: “I am happy with the way the authors have addressed my comments from the initial review.”

Thank you.

Reviewer #4 (Remarks to the Author):

1: “I was asked to fill in for Reviewer #1 of the first round of reviews. I feel that his/her points were overall addressed sufficiently and will not comment further on the replies to the other two reviewers. Instead, I would like to ask to the authors about a major additional concern I have concerning their emulation strategy:

The issue of extrapolation using the boosted tree emulator already came up several times during the first round of reviews, but never in a context that I would have expected to be most obvious: the application of an emulator trained on one model system (CESM) to another, with potentially completely different values ranges for the atmospheric forcings. This could/should be a major problem I am afraid, because it is well-known that tree models are not able to extrapolate. Even basic blog posts have discussed such issues with tree models to great length, e.g.

<http://freerangestats.info/blog/2016/12/10/extrapolation>

So, my question is how the emulator should at all be able to reproduce realistic temperature extremes (or their uncertainties) provided input of atmospheric forcings that could lie well outside its training range. For example, if the surface warming attained by 2070 is between 2.5 and 3.0 degrees across the CESM ensemble members, how would the emulator be able to predict a realistic heat extreme for another model which has warmed by only 2 degrees or even 4 degrees? Tree models should be particularly bad in these situations (even among machine learning models). The authors say in reply to Reviewer #1 on p.8 of the response to the reviewers that they have mitigated such effects by using data from the same decade for each CMIP model (i.e. the same decade as used to train CESM). However, CESM and the CMIP models will almost certainly have warmed to substantially different degrees (and of course this will likely also apply to other forcing variables). I expect that the CMIP model inputs ranges to the emulator will be substantially wider than the training range encountered in CESM. I recommend the authors demonstrate that this is not the case and otherwise check if their study is still valid/what the implications are.”

The reviewer has raised a great point regarding the extrapolation facing the machine learning method. The reviewer is correct that extrapolation is a common problem for virtually all the machine learning algorithms. We agree with the reviewer that the machine learning methods including XGBoost generally perform better in terms of interpolation than extrapolation. This is why in our original attempt we applied the emulator to the same time period of each CMIP model to avoid extrapolation. However, XGBoost is actually able to predict values outside of the training range. In a regression problem, the final prediction result depends on the previous gradient boosted trees, rather than whether it is bound or not. It is even more likely to see such out-of-bound predictions, when the feature inputs (or dimensions) of the model become large (which is the case in this study). The situation described in the blog post raised by the reviewer (i.e., the predictions will always be around the extreme value of the labels when the feature is outside the training range) is more likely to occur in a single-tree model or with one dimension, as also noted in the post. This is also evidenced by our results. If our emulator always predicts around the bound of the CESM-modeled urban temperatures, we would not have obtained a much larger range of the emulated CMIP temperatures than of CESM-modeled ones. The multi-model urban heat wave (UHW) projections would have been about the same with CESM UHW projections.

Nevertheless, we agree with the reviewer that if the new data is too far outside the training range, the XGBoost won't perform well, similar to other machine learning methods. To address the reviewer's concern, we have checked how the CESM training data covers the CMIP5 forcing data. We compared the atmospheric forcing ranges of the original training data (CESM) and CMIP5 models for the year 2061-2070 in every grid cell. Figure R1 attached below illustrates the number of CMIP5 models with their 95% forcing data covered by the original CESM training forcings. Our results suggest that the original training data, because of coming from a substantially large number of ensembles, actually cover a good range of the CMIP5 atmospheric forcing.

Figure R1. The range of forcing variables in CMIP5 models well covered by the original CESM training data (10% of the 2061-2070 data from each member). The colors indicate the number of CMIP5 models out of 17 in total with the ranges of the forcings (2.5th percentile and 97.5th percentile) covered by the original CESM training data.

Although the CMIP5 forcings are shown to be well captured by the original training data, we do think that the reviewer has provided a very good suggestion to further improve the method. In order to make the emulator “see” an even larger range of training data, instead of using 10 years of CESM Large Ensemble simulations we now use 30 years (2051-2080) – ten years before and after the prediction period – to train the emulator but predict 10 years (2061-2070) when the emulator is applied to CMIP5 models. In this way, the training data covers an even better range (Figure R2). With this updated emulator trained on 30 years of CESM simulations, we obtained quite consistent results with previous ones across all the estimates in discussion, further demonstrating the robustness and credibility of the results. These new estimates are noted and highlighted in the revised manuscript.

Figure R2. Same with Figure R1 except for 30 years of training data from CESM simulations.

We have incorporated this result in the manuscript and also noted in the text that one can choose a longer time period for training than for inference to avoid far extrapolation. We think this could potentially make the emulator strategy more flexible for other model applications, especially for the ESMs that do not have large ensemble simulations. The text modified has been reproduced below:

“We utilize the fully-coupled CESM-LE simulations as the training sets to build the urban daily temperature emulator. We randomly sample 10% of the 2006–2015 simulation outputs and 3.3% of the 2051–2080 from each member of the CESM-LE simulations within each grid cell that has an urban land unit to train the emulator for 2006-2015 and for 2061–2070 respectively. Each member then contributes to the training dataset with the same weight. With this strategy, we ensure a sufficiently large size of training data (> 3 times of using only one CESM-LE member data). Note that we selected a much longer period (30 years) of data to train the emulator for the future period (2061-2070). This attempts to avoid extrapolation for the inference stage when the emulator is applied to other CMIP5 ESMs. With a longer period of training data, the emulator could “see” a much wider range of the features (forcing fields) and label (urban temperature). Supplementary Fig. 3 demonstrates that the training set well captures the ranges of the atmospheric forcing fields in CMIP5 models.” (Line 461-471)

2: “Related to this point: I also find the range of hyperparameters optimized over not particularly comprehensive. I am further wondering why one would train on 50 estimators as well if you can afford 500, this computational expense would have been better spent on more detailed grid searches for other hyperparameters.”

Thanks for the good question and we appreciate the suggestion. We tested both 50 and 500 for the number of estimators, as we initially wanted to test a larger range. We agree with the reviewer that the computational expense could be spent on more detailed search for other hyperparameters. Therefore, we conducted a Bayesian Optimization approach over a much finer detailed search space along with the 30 years of new training data to estimate the optimal hyperparameters. We used the “BayesSearchCV” package which combines Bayesian Optimization and cross-validation to comprehensively search the hyperparameter space. The ranges that we set are as below:

- n_estimator: [10, 600]
- max_depth: [2, 7]
- learning_rate: [0.01, 1]

Based on the cross-validation with 128 iterations (~30 core hours), the algorithm concludes that the best combination is:

- n_estimators = 576
- max_depth = 6
- learning_rate = 0.088

The best combination is actually similar to the previous one obtained by grid search (n_estimator = 500, max_depth = 6, and learning_rate = 0.05).

With the new hyperparameters, the performance of the emulator is consistent with the previous one (slightly improved over the years 2006-2015).

Table R1. Emulator Performance for the year 2006-2015.

Methods	Training Data	mean of rmse	mean of mse
Grid Search	2006-2015	0.729	0.612
BayesianCV	2006-2015	0.725	0.605

Table R2. Emulator Performance for the year 2061-2070.

Methods	Training Data	mean of rmse	mean of mse
Grid Search	2061-2070	0.725	0.596
BayesianCV	2051-2080	0.743	0.624

With the new hyperparameters and 30 years of new training set, all the final results are quite consistent with the previous results, as demonstrated by only slightly different numerical values throughout the manuscript. For example, the new estimated mean UHW intensities for the four hotspot regions are 2.2 K (Great Lakes region of North America), 1.9 K (Southern Europe), 1.4 K (Central India), and 2.0 K (North China), compared with the previous estimates of 2.0 K, 2.0 K,

1.5 K, and 2.0 K respectively. Therefore, all major conclusions from the study remain the same. This broad agreement further demonstrates the robustness of the methodology and validity of the results.

We have now included this new hyperparameter optimization in the Methods section as below:

“We employed a Bayesian Optimization with 5-fold cross validation⁶⁸ to search for an optimal combination of three key hyperparameters of XGBoost including the number of gradient boosted trees (n_estimators), the maximum tree depth for base learners (max_depth), and the boosting learning rate (learning_rate). Because our entire global daily dataset is huge and thus expensive for multifold cross validation, we randomly sampled 0.1% of the entire training dataset of all the grid cells that have an urban land unit for both 2006-2015 and 2051-2080 which comprises more than 100000 data samples for the search. The search space consists of n_estimators (ranging from 10 to 600), max_depth (from 3 to 7) and learning_rate (from 0.01 to 1). Based on the average out-of-sample performance of the 5-fold cross validation after 128 iterations, the best combination from our search space is 576 (for n_estimators), 6 (for max_depth), and 0.088 (for learning_rate).”
(Line 435-444)

Minor comment:

4: “- p.5, l.110: you mean ‘on which day of year’, or ‘on which day in a series of days’? Otherwise, this really doesn’t make sense to me.”

We have now modified the text as below:

“Models project similar magnitudes as well as the frequency of temperature extremes, but do not necessarily agree on which days of the year a heat wave event occurs.” (Line 110-111)

Reviewer comments, third round

Reviewer #4 (Remarks to the Author):

While I am still concerned about extrapolation issues and thus the generalizability/applicability of the method, I do acknowledge that the authors have addressed my comments and have added to a discussion on the treatment of these concerns in their Methods section.

As a final comment I suggest that the limitation of extrapolation (which necessarily still exists) should/could also be mentioned in the main part of the paper, not just in the Methods section. It is important aspect to understand and to be aware of for future users of the method. It still makes me feel uncomfortable that we can't evaluate the method objectively on the CMIP data.

REVIEWER COMMENTS

Reviewer #4 (Remarks to the Author):

1: “While I am still concerned about extrapolation issues and thus the generalizability/applicability of the method, I do acknowledge that the authors have addressed my comments and have added to a discussion on the treatment of these concerns in their Methods section.”

Thank you.

2: “As a final comment I suggest that the limitation of extrapolation (which necessarily still exists) should/could also be mentioned in the main part of the paper, not just in the Methods section. It is important aspect to understand and to be aware of for future users of the method. It still makes me feel uncomfortable that we can't evaluate the method objectively on the CMIP data.”

We appreciate the reviewer’s suggestion. We have now included the following text in the Main text of the manuscript:

“We employ the tree-based XGBoost [66] to fit separate emulators for the present day (defined as 2006-2015) and future projected climate (2061-2070). The emulators are then applied to 17 ESMs that participated in the CMIP5 [29] to generate global multi-model projections of local urban daily maximum (T_{max}) and minimum temperatures (T_{min}) under the Representative Concentration Pathway (RCP) 8.5 scenario. In this way, the emulator essentially functions by driving the urban model in CESM with atmospheric forcings from various ESMs in the CMIP5 in a statistical way instead of a numerical way (see Methods). Note that we use the same ten years of CESM simulations (2006-2015) to train the emulator for the present day, but thirty years (2051-2081) of data to train the emulator for the future period (2061-2070). This strategy aims to minimize the extrapolation errors associated with the machine learning when the emulator is applied to other CMIP5 ESMs (see Methods).” (Line 81-91)